# IA-GGAD: Zero-shot Generalist Graph Anomaly Detection via Invariant and Affinity Learning

**Xiong Zhang**[*]
School of Software, Yunnan University
zhangxiong@stu.ynu.edu.cn

**Zhenli He**[*]
School of Software, Yunnan University
hezl@ynu.edu.cn

**Changlong Fu**
School of Software, Yunnan University
fuchanglong@stu.ynu.edu.cn

**Cheng Xie**[†]
School of Software, Yunnan University
xiecheng@ynu.edu.cn

## Abstract

Generalist Graph Anomaly Detection (GGAD) extends traditional Graph Anomaly Detection (GAD) from one-for-one to one-for-all scenarios, posing significant challenges due to Feature Space Shift (FSS) and Graph Structure Shift (GSS). This paper first formalizes these challenges and proposes quantitative metrics to measure their severity. To tackle FSS, we develop an anomaly-driven graph invariant learning module that learns domain-invariant node representations. To address GSS, a novel structure-insensitive affinity learning module is introduced, capturing cross-domain structural correspondences via affinity-based features. Our unified framework, IA-GGAD, integrates these modules, enabling anomaly prediction on unseen graphs without target-domain retraining or fine-tuning. Extensive experiments on benchmark datasets from varied domains demonstrate IA-GGAD 's superior performance, significantly outperforming state-of-the-art methods (e.g., achieving up to +12.28 % AUROC over ARC on ACM). Ablation studies further confirm the effectiveness of each proposed module. The code is available at https://github.com/kg-cc/IA-GGAD/.

## 1 Introduction

Node-level Graph Anomaly Detection (GAD) has become an essential tool for identifying suspicious entities in complex relational data. Its applications span many critical domains [1, 2, 3, 4, 5, 6, 7]. For example, in finance, anomalous transaction networks can reveal fraud or money laundering [1, 4, 8]; in social media, detecting abnormal user behavior or bot accounts is essential for maintaining platform integrity [9, 10, 11]; and in e-commerce, uncovering unusual purchase or review patterns helps identify fraudulent activity [12, 13]. Modern GAD techniques, often based on graph neural networks [14, 15, 16] or statistical models [17, 18, 19], have achieved great success on individual tasks, but training separate detectors for each graph or domain is time-consuming and costly, limiting the applicability of GAD in unseen anomaly detection environments.

In practice, organizations often manage multiple graph data sources and seek a unified anomaly detector. This motivates *Generalist Graph Anomaly Detection* (GGAD), a new and challenging setting in which a single model must detect anomalies across diverse graph domains[20, 21]. Unlike traditional GAD, which trains a dedicated detector per dataset, a generalist GAD model aims for *one model for all domains*. Such cross-domain detection is highly significant in real-world deployments;

---

[*]These authors contributed equally.

[†]Corresponding author.

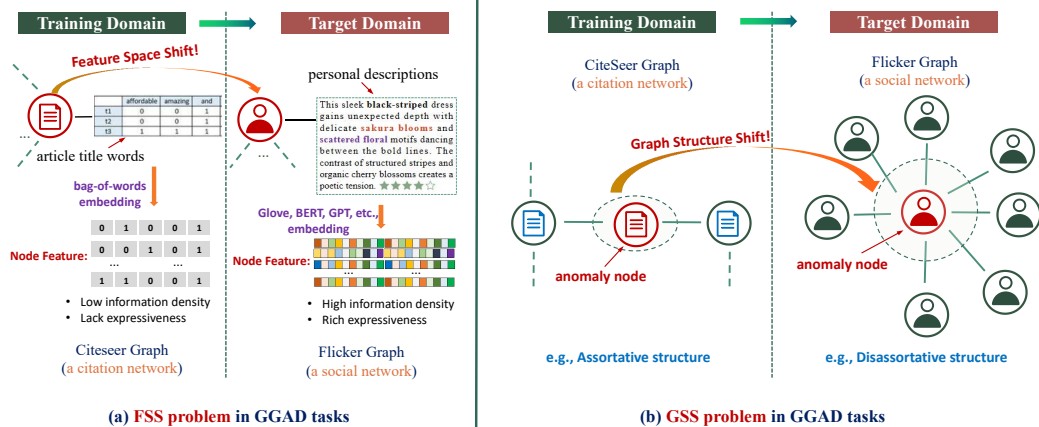

Figure 1: The motivation of the work in the GGAD scenario. (a) A case of the FSS problem between the CiteSeer and Flickr datasets. (b) A case of the GSS problem between the Citeseer and Flickr datasets.

for example, a single system could detect fraud in a financial transaction network and malicious accounts in a social network using the same model, reducing maintenance burden. However, this generalization requirement introduces substantial domain-shift challenges between different graphs.

Existing GAD approaches have not fully addressed GGAD. Traditional methods typically assume that data comes from a single graph and must be retrained or fine-tuned on each new dataset. Recently, a few works have attempted to develop generalist GAD models. For instance, ARC [20] employs an in-context learning strategy with a feature alignment module to extract cross-dataset anomaly patterns, and Unprompt [22] uses a prompt-based mechanism to unify node representations across graphs. While these methods make strides toward generalization, they still struggle to handle fundamental cross-domain shifts. Specifically, we identify two core domain-shift challenges:

- **Feature Space Shift (FSS)**: Differences in node feature distributions or semantics across domains. Nodes from different domains may differ in feature scale, dimensionality, or meaning, causing models trained on one domain to misinterpret features from another, as illustrated in Figure 1 (a).
- **Graph Structure Shift (GSS)**: Variability in graph structure across different domains. Graphs can exhibit significantly different connectivity patterns (e.g., community structures, average degrees), causing structural patterns normal in one domain to appear anomalous in another, as illustrated in Figure 1 (b).

These shifts cause existing models to misinterpret node features or structural cues when applied to new graphs. To tackle these challenges, we propose **IA-GGAD** (Invariance and Affinity Combined Graph Anomaly Detection), a novel framework that explicitly addresses both FSS and GSS. IA-GGAD combines two key components: *invariant feature learning* and *structure-insensitive affinity learning*. The invariant feature component learns node representations that capture essential anomaly-related patterns while being insensitive to domain-specific feature shifts, effectively aligning feature spaces across domains. The graph affinity component learns cross-graph structural correspondences, enabling the model to transfer anomaly cues across heterogeneous structures. These components are trained jointly in a unified architecture. Notably, IA-GGAD requires no access to any target-domain data or labels at test time for retraining or fine-tuning. Once trained on source graphs, the model can be directly applied to an unseen target graph without additional adaptation.

We evaluate IA-GGAD on benchmark datasets from diverse domains, including social networks, citation networks, and e-commerce graphs. Empirical results demonstrate that IA-GGAD achieves state-of-the-art anomaly detection performance across all tasks. It consistently outperforms existing GGAD baselines, achieving substantial improvements in AUROC. For example, on the ACM citation network, IA-GGAD improves AUROC by +12.28% compared with ARC [20]. Importantly, these gains are obtained without any target-domain retraining or labeled data, underscoring the practical utility of our zero-shot approach.

## 2 Related Work

Recent advances in anomaly detection have moved beyond task-specific models toward generalist approaches that apply to diverse domains.

**Graph Anomaly Detection.** Graph anomaly detection (GAD) aims to identify nodes or substructures in a graph that deviate from normal patterns [3, 23, 24]. Traditional GAD methods are typically trained on a single dataset and can be categorized by supervision level. Supervised GAD uses labeled anomalies (or normal nodes) to train a classifier, but acquiring anomaly labels is often infeasible in practice, such as BGNN [25], BWGNN [3], GHRN [26], and CAGAD [27]. Semi-supervised GAD[28] assumes a subset of nodes are labeled as normal. For example, Qiao *et al.*[28] propose S-GAD, a generative semi-supervised method that synthesizes artificial outliers from known normal nodes to train a one-class classifier. Unsupervised GAD operates without any labels, often using graph autoencoders or one-class objectives. DOMINANT[15] employs GCNs [14] as an autoencoder to reconstruct graph structure and node attributes and identifies anomalies via the reconstruction error. CoLA[16], based on contrastive learning, uses a discriminator to detect inconsistencies between the target node and neighbor subgraph embeddings. Similarly, TAM[29] and GCTAM[30] extend one-class deep learning to graphs by optimizing an affinity objective over GNN embeddings. These unsupervised methods achieve strong performance on individual graphs but must be retrained for each new graph, limiting their cross-domain generalization.

**Generalist Graph Anomaly Detection.** To overcome the one-model-per-dataset limitation, recent work proposes generalist graph anomaly detection frameworks. Although some GAD approaches [31, 32] can handle cross-domain scenarios, their requirement for high correlation (e.g., aligned node features) between source and target datasets limits their generalizability. Differing from those, Liu *et al.*[20] introduce ARC, a "one-for-all" GAD framework based on in-context learning. ARC aligns node features from different graphs using a learned feature-space projection, encodes residual neighborhood patterns via an ego-neighbor graph encoder, and employs a cross-attentive scoring module that compares nodes to a few-shot set of normal examples. This design allows ARC to adapt to new graphs with minimal additional data. Niu *et al.*[22] present UNPrompt, a zero-shot generalist GAD model trained on a single source graph. UNPrompt aligns the dimensionality and semantics of node attributes across graphs through coordinate-wise normalization and learns generalized neighborhood prompts so that the predictability of latent node attributes serves as a universal anomaly score. Despite their contributions, ARC and UNPrompt have limitations. ARC still requires a few target-domain normal examples at inference, and its learned alignment may not eliminate all domain shifts. UNPrompt relies on consistent attribute semantics and a single-source training setup, which can limit its applicability when graphs vary widely in feature space or structure.

## 3 Problem Statement

This section presents the formal problem definition and highlights the key challenges that our framework aims to address.

**Preliminaries.** Let us define an attributed graph as $\mathcal{G} = (\mathcal{V}, \mathbf{A}, \mathbf{X})$, where $\mathcal{V} = \{v_1, \ldots, v_N\}$ is the set of $N$ nodes, $\mathbf{X} \in \mathbb{R}^{N \times d} = \{x_1, \cdots, x_N\}$ is the node feature matrix, and $\mathbf{A} \in \{0, 1\}^{N \times N}$ is the adjacency matrix such that $\mathbf{A}_{ij} = 1$ indicates an edge between $v_i$ and $v_j$. Each node $v_i$ is associated with a feature vector $x_i \in \mathbb{R}^d$.

**Task Definition.** In the GGAD setting, we aim to learn a generalist graph anomaly detection model from a collection of labeled source-domain graphs $\mathcal{T}_{\text{train}} = \{\mathcal{D}_{\text{train}}^{(1)}, \ldots, \mathcal{D}_{\text{train}}^{(n_o)}\}$, and generalize it to a disjoint set of target-domain graphs $\mathcal{T}_{\text{test}} = \{\mathcal{D}_{\text{test}}^{(1)}, \ldots, \mathcal{D}_{\text{test}}^{(n_t)}\}$ without access to any target-domain labels or retraining, where $n_o$ and $n_t$ denote the number of source-domain training datasets and target-domain testing datasets, respectively. Each $\mathcal{D}^{(i)} = (\mathcal{G}^{(i)}, \mathbf{y}^{(i)})$ is a labeled dataset from an arbitrary domain. Unlike traditional GAD methods that build graph-specific detectors, our goal is to develop a single generalist graph anomaly detector that can effectively identify anomalies across diverse graph domains with varying semantics and structures.

**Key Challenges.** GGAD introduces significant challenges due to inherent discrepancies across graph domains. Through empirical investigation and prior studies, we identify two principal sources of domain shift that significantly degrade generalization performance: Feature Space Shift (FSS) and

Graph Structure Shift (GSS). To rigorously characterize and quantify the severity of these shifts, we introduce two mathematical formulations:

*(1) Feature Space Shift (FSS).* Given the node feature matrices $\mathbf{X}^o$ and $\mathbf{X}^t$ from source and target domains, respectively, we use Maximum Mean Discrepancy (MMD) to measure the distance between their feature distributions in a reproducing kernel Hilbert space (RKHS). Specifically,

$$\text{FSS}(\mathbf{X}^o, \mathbf{X}^t) = \frac{1}{N_o^2} \sum_{i,j} \langle \phi(x_i^o), \phi(x_j^o) \rangle + \frac{1}{N_t^2} \sum_{i,j} \langle \phi(x_i^t), \phi(x_j^t) \rangle - \frac{2}{N_o N_t} \sum_{i,j} \langle \phi(x_i^o), \phi(x_j^t) \rangle, \tag{1}$$

where $\phi(\cdot)$ is an RKHS mapping function, referring to Eq.(3), and $N_o$, $N_t$ denote the number of nodes in source and target graphs, respectively. A larger FSS score reflects greater feature misalignment, leading to unreliable inference across domains.

*(2) Graph Structure Shift (GSS).* Similarly, we quantify structural discrepancies using degree-derived structural vectors and a Gaussian kernel $f(x,y) = \exp(-\|x - y\|^2/2\sigma^2)$. Given adjacency matrices $\mathbf{A}^o$ and $\mathbf{A}^t$ from source and target graphs, GSS is computed as:

$$\begin{aligned}
\text{GSS}(\mathbf{A}^o, \mathbf{A}^t) = &\frac{1}{N_o^2} \sum_{i,j} f([\mathbf{A}^o \cdot \mathbf{A}^o]_i, [\mathbf{A}^o \cdot \mathbf{A}^o]_j) + \frac{1}{N_t^2} \sum_{i,j} f([\mathbf{A}^t \cdot \mathbf{A}^t]_i, [\mathbf{A}^t \cdot \mathbf{A}^t]_j) \\
&- \frac{2}{N_o N_t} \sum_{i,j} f([\mathbf{A}^o \cdot \mathbf{A}^o]_i, [\mathbf{A}^t \cdot \mathbf{A}^t]_j).
\end{aligned} \tag{2}$$

This metric reflects the misalignment in higher-order structural properties such as connectivity profiles and neighborhood distributions. A high GSS score indicates a greater structure divergence between source and target graphs.

**Empirical Evidence.** To further investigate the FSS and GSS challenges, we conduct a quantitative analysis of both FSS and GSS, with detailed results provided in Appendix B.

# 4   Method

To address the twin challenges of FSS and GSS identified in Section 3, we introduce IA-GGAD, an end-to-end, zero-shot framework that learns to detect anomalies on unseen graphs after training on multiple, diverse source domains. Figure 2 gives an overview of IA-GGAD. IA-GGAD is composed of four tightly coupled modules: **(1) Invariant feature pool construction** (Section 4.1): align various node representations and extract a shared invariant feature pool composed of domain-invariant prototypes; **(2) Graph-invariant representation learning** (Section 4.2): embeds node representations using the shared invariant feature pool ensuring that normal and abnormal patterns are consistently separated across domains; **(3) Structure-insensitive affinity learning** (Section 4.3): learns affinity-based node representations to enable structure-insensitive cross-graph affinity scoring; **(4) Joint anomaly scoring and prediction** (Section 4.4): fuses semantic and structural evidence to produce a final, domain-agnostic anomaly score and predictions.

## 4.1   Invariant Feature Pool Construction

The first step toward cross-domain generalization is to eliminate feature divergence while retaining critical anomaly-related and normal patterns.

**Feature alignment.** Empirical evidence in Table 5 shows that node attributes differ widely in scale, dimensionality, and semantics across datasets. We therefore project every raw feature matrix $\mathbf{X} \in \mathbb{R}^{N \times d_i}$ to a *shared* latent dimension $d_u$:

$$\bar{\mathbf{X}} = \text{proj}(\mathbf{X}) \in \mathbb{R}^{N \times d_u},$$

where $\text{proj}(\cdot)$ is a learnable linear layer that can be initialized with classical dimensionality-reduction techniques such as PCA [33]. This simple yet effective step standardizes the input space for the subsequent graph encoder.

**Shared graph encoder.** Following alignment, we feed $\bar{\mathbf{X}}$ into a stack of parameter-shared Graph Convolutional Networks (GCNs) [14] to obtain multihop node embeddings:

$$\mathbf{Z}^{[1]} = \sigma\left(\mathbf{D}^{-\frac{1}{2}} \mathbf{A} \mathbf{D}^{-\frac{1}{2}} \bar{\mathbf{X}} \mathbf{W}^{[1]}\right), \ldots, \mathbf{Z}^{[\ell]} = \sigma\left(\mathbf{D}^{-\frac{1}{2}} \mathbf{A} \mathbf{D}^{-\frac{1}{2}} \mathbf{Z}^{[\ell-1]} \mathbf{W}^{[\ell]}\right) = \left\{z_1^{[\ell]}, \ldots, z_N^{[\ell]}\right\}, \tag{3}$$

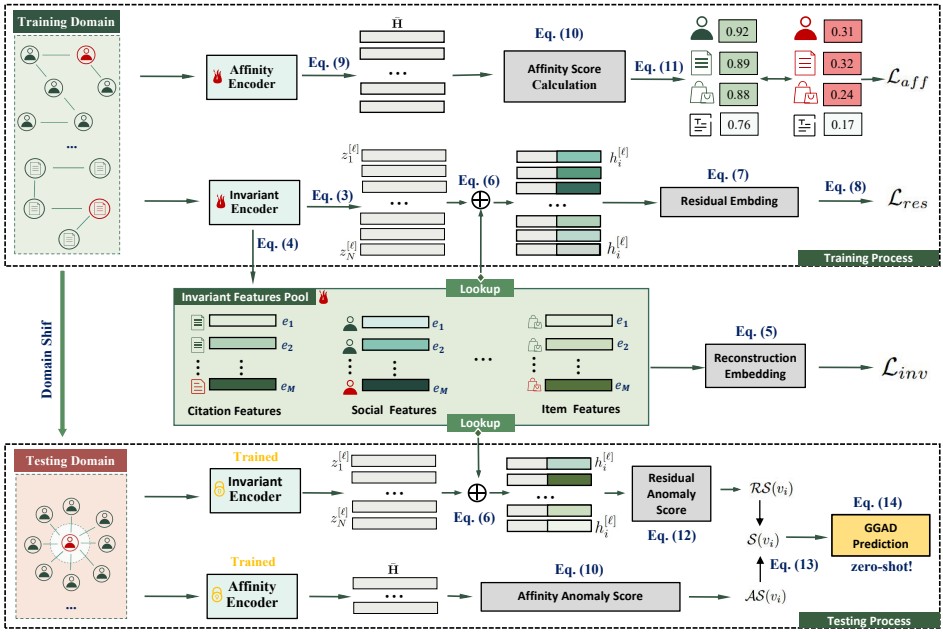

Figure 2: Overview of IA-GGAD.

where $\mathbf{A}$ is the adjacency matrix with self-loops, $\mathbf{D}$ its degree matrix, $\mathbf{W}^{[\ell]}$ are learnable weights, $\sigma(\cdot)$ is an activation function, and $\mathbf{Z}^{[\ell]}$ is the transformed representation matrix at the $\ell$-th layer.

**Invariant feature extraction.** To make the representations immune to feature-space shift, we adopt a modified Vector-Quantized VAE (VQ-VAE) [34]. Each source graph contributes to a *shared* invariant features pool $E = \{e_1, \ldots, e_M\} \subset \mathbb{R}^{M \times d_I}$ whose entries serve as domain-invariant prototypes. The invariant features pool is initialized from sampled embeddings and refined by the standard assignment–update routine:

$$e_i^{(l-1)} = \frac{1}{|N_i|} \sum_j^{N_i} z_j^{[\ell](l)}, \quad N_i = \underset{j \in \{1, \cdots, N\}}{\operatorname{argmin-}k} (||z_j^{[\ell](l)} - e_i^{(l-1)}||_2), \quad e_i^{(0)} = z_i^{[\ell](1)}, \tag{4}$$

where $e_i^{(l-1)}$ denotes the result of the $i$-th invariant feature prototype at the $(l-1)$-th iteration. In each iteration, $e_i$ is updated as the mean of its $k$ nearest node embeddings $z_i$. After convergence, a lightweight decoder reconstructs the aligned features $\bar{\mathbf{x}}_i$ from the selected prototype, and the entire module is trained with the composite loss:

$$\mathcal{L}_{inv} = \frac{1}{N} \sum_{i=1}^N \left(1 - \frac{\bar{\mathbf{x}}_i \cdot \hat{\mathbf{x}}_i}{\|\bar{\mathbf{x}}_i\| \cdot \|\hat{\mathbf{x}}_i\|}\right)^\gamma + \frac{1}{N} \sum_{i=1}^N \|\operatorname{sg}[z_i] - e_i\|_2^2 + \frac{\eta}{N} \sum_{i=1}^N \|\operatorname{sg}[e_i] - z_i\|_2^2. \tag{5}$$

The composite loss comprises three intuitive terms. (i) *Reconstruction.* The scaled cosine error $\left(1 - \frac{\bar{\mathbf{x}}_i \hat{\mathbf{x}}_i}{\|\bar{\mathbf{x}}_i\|\|\hat{\mathbf{x}}_i\|}\right)^\gamma$ $(\gamma \geq 1)$ enforces faithful recovery of aligned attributes, encouraging the encoder to retain informative semantics. (ii) *VQ update.* The vector-quantization loss $\|\operatorname{sg}[z_i] - e_i\|_2^2$ pushes each invariant features pool entry $e_i$ toward its assigned encoder output $z_i$, progressively refining the invariant prototypes. (iii) *Commitment.* The commitment term $\|\operatorname{sg}[e_i] - z_i\|_2^2$ penalizes large deviations of the encoder from the selected prototype, stabilizing training. A single hyper-parameter $\eta$ (empirically 0.25) balances invariant features pool adaptation and encoder commitment, while the stop-gradient operator $\operatorname{sg}[\cdot]$ prevents back-propagation through the detached branch.

## 4.2 Graph-Invariant Representation Learning

While conventional GNNs consume only the $\ell$-layer output embeddings, IA-GGAD enriches every node representation with explicit *invariant semantics* extracted from the invariant features pool $E$.

This fusion yields embeddings that are simultaneously sensitive to local graph context and robust to cross-domain feature shifts.

**+Invariant-feature guided embedding fusion.** For each GCN layer $\ell$, we retrieve the $k$ invariant features pool vectors whose cosine similarity with $z_i^{[\ell]}$ is highest. Averaging these invariant features and concatenating with $z_i^{[\ell]}$ gives an *invariant-aware* embedding:

$$h_i^{[\ell]} = \texttt{concat}(z_i^{[\ell]}, \frac{1}{|T_i|}\sum_{j \in T_i} e_j), \quad T_i = \operatorname*{argmax-}_{j \in \{1,...,N\}}k\left(\frac{z_i^{[\ell]} \cdot \mathbf{e}_j}{|z_i^{[\ell]}| \cdot |\mathbf{e}_j|}\right), \tag{6}$$

where $h_i^{[\ell]} \in \mathbb{R}^{d_e}$ is the output embedding matrix with invirant code emebding, $T_i$ is a set of $k$ invariant feature emebding that are closest to the current node representation $z_i^{[\ell]}$.

**Multi-hop residual encoding.** To capture how a node's representation *drifts* across message-passing layers, we take successive differences with the 1-hop baseline $h_i^{[1]}$ and concatenate them:

$$r_i = (h_i^{[2]} - h_i^{[1]}) \, || \, (h_i^{[3]} - h_i^{[1]}) \, || \, \cdots \, || \, (h_i^{[\ell]} - h_i^{[1]}), \tag{7}$$

where $||$ denotes vector concatenation. The resulting vector $r_i$ encodes multi-scale deviations that are highly informative for anomaly assessment.

**Residual similarity loss.** Given source-domain labels, we encourage residuals of *normal* nodes to cluster while pushing those of *abnormal* nodes away beyond a margin $\epsilon$:

$$\mathcal{L}_{res} = \sum_t^{N_t^+}\sum_i^{N_i^+}(1 - \frac{r_t^+ \cdot r_i^+}{||r_t^+|| \cdot ||r_i^+||}) + \sum_t^{N_t^+}\sum_j^{N_j^-}\texttt{max}(0, \frac{r_t^+ \cdot r_j^-}{||r_t^+|| \cdot ||r_j^-||} - \epsilon), \tag{8}$$

where $r_t^+$ and $r_j^-$ denote residuals of normal and anomalous nodes, respectively.

**Joint optimization.** The invariant feature pool, GCN encoder, and residual module are optimized *jointly* with $\mathcal{L} = \mathcal{L}_{\text{inv}} + \mathcal{L}_{res}$, such that semantic invariance (Section 4.1) and residual discriminability reinforce each other throughout training. As a result, the learned node embeddings exhibit strong generalization ability to unseen graphs, even in the presence of severe feature space shift (FSS).

## 4.3 Structure-Insensitive Affinity Learning

FSS-robust embeddings alone are insufficient when two graphs differ markedly in structure. To explicitly cope with GSS, we introduce a **Graph Affinity Encoder** (GAE) that learns a homophily-driven *local affinity score*. Normal nodes are expected to exhibit *high* affinity with their neighbours, whereas anomalous nodes break this pattern and thus receive *low* affinity. Optimising this contrast yields a structure-aware signal that complements the invariant semantics learned in Section 4.2.

**GNN-based node projection.** GAE first maps each node to a latent space that captures neighbourhood context. For computational efficiency and fair comparison with prior work, we employ a single Graph Convolutional Network (GCN) layer [14]:

$$\bar{\mathbf{H}} = \sigma(\mathbf{D}^{-\frac{1}{2}}\mathbf{A}\mathbf{D}^{-\frac{1}{2}}\bar{\mathbf{X}}\mathbf{W}) = \{\bar{h}_1, \cdots, \bar{h}_N\}, \tag{9}$$

where $\bar{\mathbf{X}}$ is the feature-aligned input from Section 4.1, $\mathbf{A}$ the adjacency matrix with self-loops, and $\sigma(\cdot)$ an activation function.

**Local affinity score.** Given $\bar{\mathbf{H}}$, we quantify how well node $v_i$ conforms to its immediate neighbourhood $\mathcal{N}(v_i)$ via the *average cosine similarity*:

$$\mathcal{AS}(v_i) = \frac{1}{|\mathcal{N}(v_i)|}\sum_{v_j \in \mathcal{N}(v_i)} \frac{\bar{h}_i \cdot \bar{h}_j}{|\bar{h}_i||\bar{h}_j|}. \tag{10}$$

A high value indicates strong homophily—typical for normal nodes; low values signal structural irregularities, hinting at anomalies.

**Unsupervised affinity maximisation.** We train GAE in an unsupervised manner by *maximising* each node's local affinity:

$$\mathcal{L}_{aff} = \min_{\Theta} \Big( -\sum_{i=1}^{N} \mathcal{AS}(v_i) \Big). \tag{11}$$

Optimizing (11) encourages neighbourhood-coherent representations for the vast majority of normal nodes, implicitly relegating anomalous nodes to the low-affinity tail.

### 4.4 Joint Anomaly Scoring and Prediction

Having trained the three upstream modules on the source-domain set $\mathcal{T}_{\text{train}}$, IA-GGAD performs inference on *any* unseen graph by issuing two complementary anomaly signals that mirror the twin challenges of FSS and GSS.

**Residual Score $\mathcal{RS}$.** The residual embeddings $r_i$ defined in Eq. (7) faithfully encode feature-space deviations that may arise from FSS. We measure how isolated a node is within this residual space via the mean-squared distance to $n_k$ random sample residual spaces:

$$\mathcal{RS}(v_i) = \frac{1}{n_k} \sum_{j}^{n_k} \|r_i - r_j\|^2, \tag{12}$$

where $r_j$ are residuals from the *test* graph obtained with the frozen encoder.

**Local Affinity Score $\mathcal{AS}$.** Complementing $\mathcal{RS}$, the affinity score in Eq. (10) probes structural homophily and is thus sensitive to GSS-induced anomalies.

**Weighted Fusion.** Because the two scores live on different scales, we blend them with a weighting factor $\lambda \in [0, 1]$:

$$\mathcal{S}(v_i) = (1 - \lambda)\,\mathcal{RS}(v_i) \;+\; \lambda\big(1 - \mathcal{AS}(v_i)\big). \tag{13}$$

Intuitively, a node is deemed suspicious if it shows *either* a large residual dispersion (semantic oddity) *or* a weak local affinity (structural oddity).

**Adaptive thresholding.** To convert the anomaly score $\mathcal{S}(v_i)$ into a binary prediction, we adopt the data-driven rule in Eq. (14). The optimal threshold $\tau^*$ maximises the separation between normal and anomaly sets, after which nodes with $\mathcal{S}(v_i) \geq \tau^*$ are labelled as anomalies.

$$\begin{cases} \tau^* = \arg\max\limits_{\tau \in \{\mathcal{S}(v) | v \in \mathcal{N}\}} \left[ \dfrac{1}{|\mathcal{N}^+|} \sum\limits_{v_i \in \mathcal{N}^+} \mathbb{I}\left(\mathcal{S}(v_i) \geq \tau\right) - \dfrac{1}{|\mathcal{N}^-|} \sum\limits_{v_j \in \mathcal{N}^-} \mathbb{I}\left(\mathcal{S}(v_j) \geq \tau\right) \right] \\ \hat{y}_i = \begin{cases} 1, & \text{if}\,\mathcal{S}(v_i) \geq \tau^* \\ 0, & \text{if}\,\mathcal{S}(v_i) < \tau^* \end{cases} \end{cases} \tag{14}$$

**Discussion.** The weighted fusion couples the strengths of semantic (FSS-oriented) and structural (GSS-oriented) cues, while the adaptive threshold obviates manual calibration on each new graph. Together, these choices complete an *end-to-end* zero-shot pipeline whose predictions remain reliable across dramatic domain shifts. Detailed algorithmic description and complexity analysis of IA-GGAD can be found in Appendix C.

## 5 Experiments

### 5.1 Experimental Setup

**Dataset configuration.** Following ARC [20], we create a deliberately shifted source/target split. The training set is

$\mathcal{T}_{\text{train}} = \{PubMed, Flickr, Questions, YelpChi\}$, while the zero-shot test set is $\mathcal{T}_{\text{test}} = \{ACM, Facebook, Amazon, Cora, CiteSeer, BlogCatalog, Reddit, Weibo\}$. All graphs contain injected or naturally occurring anomalies [35, 16, 29], providing a realistic benchmark for generalization.

**Baselines.** We compare IA-GGAD with sixteen strong competitors: six *supervised* GNNs (GCN [14], GAT [36], BGNN [25], BWGNN [3], GHRN [26], CAGAD [27]), one *semi-supervised* model (S-GAD [28]), five *unsupervised* methods (DOMINANT [15], CoLA [16], HCM-A [37], TAM [29], GCTAM [30]), and the two state-of-the-art *generalist* approaches ARC [20] and UNPrompt [22]. Implementation details for all baselines are provided in Appendix F.

**Evaluation protocol.** We report AUROC and AUPRC averaged over five random seeds, together with standard deviations [35, 38]. Each method, baselines and IA-GGAD alike, is trained once on $\mathcal{T}_{\text{train}}$ and then evaluated on every target graph in a *pretrain only* manner. Feature dimensions are aligned by inserting either a learnable or a random projection adapter before the input layer. Hyperparameters are selected via random search on the training split and remain fixed across target graphs, with no dataset-specific tuning. For IA-GGAD, the sample count is set to $n_k = 10$ unless specified. *Further implementation details can be found in Appendix G.*

Table 1: Anomaly detection performance in terms of AUROC (in percent, mean±std). Highlighted are the results ranked first, second, and third.

| Method | ACM | Facebook | Amazon | Cora | CiteSeer | BlogCatalog | Reddit | Weibo | Rank |
|---|---|---|---|---|---|---|---|---|---|
| | | | | GAD methods | | | | | |
| GCN(2017) | 60.49±9.65 | 29.51±4.86 | 46.63±3.47 | 59.64±8.30 | 60.27±8.11 | 56.19±6.39 | 50.43±4.41 | 76.64±17.69 | 9.00 |
| GAT(2018) | 48.79±2.73 | 51.88±2.16 | 50.52±17.22 | 50.06±2.65 | 51.59±3.49 | 50.40±2.80 | 51.78±4.04 | 53.06±7.48 | 10.37 |
| BGNN(2021) | 44.0±13.69 | 54.74±25.29 | 52.26±3.31 | 42.45±11.57 | 42.32±11.82 | 47.67±8.52 | 50.27±3.84 | 32.75±35.35 | 11.87 |
| BWGNN(2022) | 67.59±0.70 | 45.84±4.97 | 55.26±16.95 | 54.06±3.27 | 52.61±2.88 | 56.34±1.21 | 48.97±5.74 | 53.38±1.61 | 9.13 |
| GHRN(2023) | 55.65±6.37 | 44.81±8.06 | 49.48±17.13 | 59.89±6.57 | 56.04±9.19 | 57.64±3.48 | 46.22±2.33 | 51.87±14.18 | 9.87 |
| DOMINANT(19) | 70.08±2.34 | 51.01±0.78 | 48.94±2.69 | 66.53±1.15 | 69.47±2.02 | 74.25±0.65 | 50.05±4.92 | 92.88±0.32 | 6.12 |
| CoLA(2021) | 66.85±4.43 | 12.99±11.68 | 47.40±7.97 | 63.29±8.88 | 62.84±9.52 | 50.04±3.25 | 52.81±6.69 | 16.27±5.64 | 10.00 |
| HCM-A(2022) | 53.70±4.64 | 35.44±13.97 | 43.99±0.72 | 54.28±4.73 | 48.12±6.80 | 55.31±0.57 | 48.79±2.75 | 65.52±12.58 | 11.50 |
| TAM(2023) | 74.43±1.59 | 65.88±6.66 | 56.06±2.19 | 62.02±2.39 | 72.27±0.83 | 49.86±0.73 | 55.43±0.33 | 71.54±0.18 | 5.25 |
| CAGAD(2024)* | 39.80±9.91 | 45.84±4.97 | 46.06±0.75 | 50.11±3.41 | 50.11±3.41 | 49.84±12.37 | 54.57±3.89 | 58.99±3.42 | 11.75 |
| S-GAD(2024)* | 37.47±2.68 | 55.89±8.99 | 53.11±4.92 | 39.44±5.41 | 38.18±4.21 | 50.70±7.34 | 55.39±0.44 | 65.73±3.35 | 10.00 |
| GCTAM(2025)* | 81.21±0.13 | 69.57±1.41 | 55.74±0.60 | 58.78±2.17 | 70.31±1.77 | 67.60±0.77 | 59.32±0.73 | 70.61±0.10 | 4.37 |
| | | | | GGAD methods | | | | | |
| UNPrompt(2024)* | 69.91±1.28 | 55.27±6.90 | 56.02±11.69 | 54.31±1.50 | 49.80±3.12 | 68.36±0.40 | 59.18±1.44 | 45.56±3.75 | 7.12 |
| ARC(2024) | 79.88±0.28 | 67.56±1.60 | 80.67±1.81 | 87.45±0.74 | 90.95±0.59 | 74.76±0.06 | 60.04±0.69 | 88.85±0.14 | 2.38 |
| **IA-GGAD (ours)** | **93.49±0.57** | **80.03±1.09** | **83.78±2.76** | **88.68±0.53** | **91.83±0.43** | **75.28±0.21** | **60.29±1.91** | 91.18±0.22 | **1.12** |
| Δ | ↑12.28 | ↑10.46 | ↑3.11 | ↑1.23 | ↑0.88 | ↑0.52 | ↑0.25 | ↓1.70 | |

Δ represents the improvement (↑) or degradation (↓) compared to the current best baseline method. Rank indicates the average ranking over 8 datasets.

* represents reproduced results, others are reported in ARC[20].

## 5.2 Main Results

Table 1 summarises AUROC performance across fourteen benchmarks. IA-GGAD achieves the **top rank on 7/8** datasets, with especially large gains of **+12.28%** on ACM and **+10.46%** on Facebook. These results confirm that our joint handling of FSS and GSS yields robust zero-shot generalisation. Most strikingly, IA-GGAD outperforms the strongest baselines on Amazon (+3.11%), Cora (+1.23%), and CiteSeer (+0.88%), demonstrating consistent improvements even on highly diverse graphs. Traditional supervised GAD methods (e.g. GCN, GAT, BGNN, GHRN) fall near chance without retraining, and unsupervised approaches like DOMINANT and TAM perform well only when their implicit homophily assumptions hold. The sole exception is Weibo, where DOMINANT's reconstruction-based detector attains 92.88% versus our 91.18% (**–1.70%**), which remains a very high score. We attribute this to Weibo's extremely strong local homophily and low attribute variance, which favour autoencoder reconstruction errors over invariant-feature alignment. In contrast, IA-GGAD 's invariant pool and affinity encoder balance semantic and structural cues, leading to more stable performance across weaker-homophily graphs. Finally, IA-GGAD 's mean rank of **1.12** far surpasses ARC (2.38) and GCTAM (4.37), and its low standard deviations (all <3%) underscore stable behaviour across random seeds. Full AUPRC results appear in Appendix H.1.

## 5.3 Ablation Study

To disentangle the impact of each module, we evaluate three variants: (i) **w/o I&A**—backbone only; (ii) **w/I**—backbone plus invariant feature pool; (iii) **w/A**—backbone plus graph affinity encoder. Table 2 shows that invariant features chiefly improve FSS-dominated graphs (e.g. CiteSeer, Cora), while affinity

Table 2: The evaluation of invariant feature(I) and graph affinity(A)

| Datasets | w/o I&A | w/I | w/A | Ours |
|---|---|---|---|---|
| ACM | 78.59 | 79.86 | 91.78 | **93.49** |
| Facebook | 67.73 | 68.02 | 78.94 | **80.03** |
| Amazon | 79.47 | 80.85 | 82.76 | **83.78** |
| Cora | 87.10 | 88.14 | 87.69 | **88.68** |
| CiteSeer | 90.43 | 91.37 | 91.31 | **91.83** |
| BlogCatalog | 74.03 | 74.61 | 74.41 | **75.28** |
| Reddit | 58.31 | 59.89 | 59.21 | **60.29** |
| Weibo | 86.35 | 88.63 | 88.93 | **91.18** |

modelling is crucial for GSS-heavy graphs (ACM, Facebook, Amazon). Combining both yields the full model, outperforming every variant on all datasets.

## 5.4 Hyper-parameter Sensitivity

Figure 3 studies the weighting factor $\lambda$ in Eq. (13). On both ACM and Facebook, performance peaks at $\lambda = 0.9$, confirming that a modest contribution from affinity scores complements the residual signal without amplifying noise. Beyond this point, AUROC and AUPRC decline, underscoring the need for balanced fusion. Additional sensitivity plots are provided in Appendix 4.

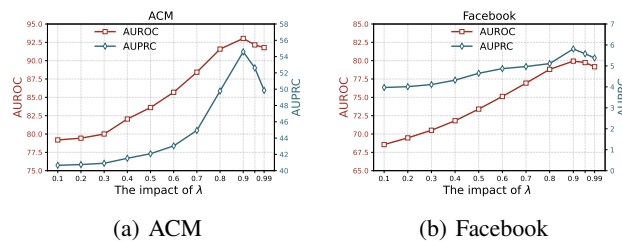

|  (a) ACM  |  (b) Facebook  |

Figure 3: Weighting Factor $\lambda$ vs AUROC and AUPRC

## 5.5 Impact of FSS and GSS

Table 3 pairs four targets with a common source (PubMed) and reports the quantitative shift measures from Eq. (1)–(2). ACM and Facebook exhibit high GSS but negligible FSS; CiteSeer and Cora show the opposite pattern. GCTAM excels under GSS but degrades under FSS; ARC shows the reverse. By contrast, IA-GGAD outperforms both baselines in *all* scenarios, confirming that the dual design—semantic invariance for FSS, affinity learning for GSS—generalises robustly across heterogeneous graphs.

Table 3: Quantitative analysis of FSS and GSS

|  | GSS Dominated | | FSS Dominated | |
|---|---|---|---|---|
| Source Dataset | Pubmed | | | |
| Target Dataset | ACM | Facebook | CiteSeer | Cora |
| FSS Score, Eq.(1) | 0.01 | 0.31 | 0.70 | 0.68 |
| GSS Score, Eq.(2) | 0.77 | 0.64 | 0.03 | 0.31 |
| GCTAM(2025) * | 81.21 | 69.57 | 70.31 | 58.78 |
| ARC(2024) ** | 79.88 | 67.56 | 90.95 | 87.45 |
| **IA-GGAD (ours)** | **93.49** | **80.03** | **91.83** | **88.68** |

\* GCTAM (SOTA for GAD) is sensitive to FSS.
\*\* ARC (SOTA for GGAD) is sensitive to GSS.

## 5.6 Performance on Large Datasets

To evaluate the scalability of IA-GGAD, we test it on large-scale financial networks with real anomalies, including DGraph [39], T-Finance [3], and Elliptic [40]. We compare IA-GGAD with recent state-of-the-art methods such as CONSISGAD [41], SmoothGNN [42], AnomalyGFM [43], and SpaceGNN [44]. IA-GGAD consistently achieves the best performance. On Elliptic, it reaches an AU-ROC of 74.24, outperforming the second-best (SpaceGNN, 57.43) by 16.81 %. On T-Finance, it scores 75.41, surpassing the next-best (AnomalyGFM, 67.57) by 11.75 %. On DGraph-Fin, where many baselines fail due to memory constraints, IA-GGAD still attains 54.39. These results demonstrate its robustness and scalability on large and complex financial graphs.

Table 4: Results (AUROC) on large datasets.

| Method | Elliptic | T-Finance | DGraph-Fin |
|---|---|---|---|
| DOMINANT (2019) | 24.80 | 59.89 | OOM |
| CoLA (2021) | 38.72 | 51.34 | OOM |
| TAM (2023) | 22.31 | 39.42 | OOM |
| CONSISGAD (2024) | 48.36 | 53.56 | 51.76 |
| SmoothGNN (2025) | 56.23 | 39.79 | 52.71 |
| SpaceGNN (2025) | 57.43 | 57.43 | 30.04 |
| GCTAM (2025) | 24.56 | 40.74 | OOM |
| ARC (2024) | 26.40 | 64.10 | 47.46 |
| UNPrompt (2024) | 41.76 | 23.86 | 52.69 |
| AnomalyGFM (2025) | OOM | 67.57 | OOM |
| **IA-GGAD (Ours)** | **74.24** | **75.41** | **54.39** |

## 6 Limitations

Although IA-GGAD substantially advances *zero-shot* graph anomaly detection, several limitations remain. **(1) Dependence on informative node attributes.** The invariant-feature module assumes moderately descriptive features; on purely structural graphs or graphs with sparse or noisy attributes, its FSS-mitigation effect can disappear and performance degrades. **(2) Homogeneity assumption in affinity learning.** The affinity encoder is based on the homogeneity assumption, which does

not directly apply to heterogeneous graphs; however, by employing meta-path, IA-GGAD can be extended to heterogeneous graph scenarios. **(3) Static structures.** Our evaluation covers only static graphs; dynamic (time-varying) graphs remain out of scope and likely require substantive extensions. **(4) Limited scale analysis.** Scalability to million-scale graphs is untested, and both the $k$-NN prototype retrieval and global affinity objective may need approximate or mini-batch variants. **(5) Hyper-parameter sensitivity.** Results depend on weighting factor $\lambda$; a comprehensive robustness study is left to future work.

## 7  Conclusion

We presented IA-GGAD, an end-to-end zero-shot framework for **GGAD** that detects anomalies on unseen graphs without retraining. By formalising and quantifying *Feature Space Shift* (FSS) and *Graph Structure Shift* (GSS), we pinpointed the key barriers to cross-domain generalisation. IA-GGAD counters them with two lightweight modules: an *anomaly-driven invariant feature pool* to mitigate FSS and a *graph affinity encoder* to withstand GSS. Across eight real-world datasets, IA-GGAD outperforms fourteen strong baselines—including ARC and UNPrompt—achieving up to +12.28% AUROC on ACM and +10.46% AUROC on Facebook, thereby setting a new benchmark for zero-shot graph anomaly detection.

**Future work.** Although treating FSS and GSS separately proves effective, a unified representation that jointly normalizes feature semantics and structural patterns could further improve robustness. In addition, we plan to extend IA-GGAD to dynamic graphs and to graphs with heterogeneous node and edge types, which would broaden its applicability in real-world scenarios.

## Acknowledgments and Disclosure of Funding

This paper is the result of the research project funded by the National Natural Foundation of China (Grant No. 62106216, 62362068, and 62162064) and the Open Foundation of Yunnan Key Laboratory of Software Engineering under Grant No.2023SE104 and No.2023SE208.

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

# A  Theoretical Foundations of FSS and GSS

This appendix provides a rigorous foundation for **FSS** and **GSS** as defined in the main text. We formalize the kernels underlying Eqs. (1)–(2), address well-definedness across domains of different sizes, establish an additive-kernel decomposition, and present a domain-adaptation bound under explicit assumptions.

## A.1  Notation and Background

**RKHS and kernels.** A Hilbert space $(\mathcal{H}, \langle \cdot, \cdot \rangle)$ is an RKHS for a positive-definite kernel $k$ if there exists a feature map $\phi$ with $k(u, v) = \langle \phi(u), \phi(v) \rangle$.

**Population and empirical MMD.** For distributions $P, Q$ over a common space $\mathcal{Z}$, the (population) MMD is

$$\mathrm{MMD}_k(P, Q) = \| \mu_k(P) - \mu_k(Q) \|_{\mathcal{H}_k}, \quad \mu_k(P) = \mathbb{E}_{z \sim P}[\phi(z)].$$

Given i.i.d. samples $\mathcal{S} = \{s_i\}_{i=1}^{n_S} \sim P$ and $\mathcal{T} = \{t_j\}_{j=1}^{n_T} \sim Q$, the biased empirical estimator is

$$\widehat{\mathrm{MMD}}_k^2(\mathcal{S}, \mathcal{T}) = \tfrac{1}{n_S^2} \sum_{i,i'} k(s_i, s_{i'}) + \tfrac{1}{n_T^2} \sum_{j,j'} k(t_j, t_{j'}) - \tfrac{2}{n_S n_T} \sum_{i,j} k(s_i, t_j). \tag{A.1}$$

Assume throughout that the kernel is bounded, $\sup_z k(z, z) \leq \kappa^2$, and that samples are drawn i.i.d. when invoking concentration bounds. Then $\left| \widehat{\mathrm{MMD}}_k(P, Q) - \mathrm{MMD}_k(P, Q) \right| = O_p\big( \sqrt{1/n_S + 1/n_T} \big)$ [45].

## A.2  Formalizing the Kernels Behind Eqs. (1)–(2)

**FSS kernel consistent with Eq. (1).** Let $G$ denote a graph domain and let $\phi_G : \mathbb{R}^d \to \mathbb{R}^h$ be the shared graph encoder *with a single set of shared parameters held fixed when computing MMD*. Define a single kernel on graph-tagged inputs

$$k_f\big((x, G), (x', G')\big) := \big\langle \phi_G(x), \phi_{G'}(x') \big\rangle.$$

This is positive-definite since it is an inner product in a common feature space via the unified feature map $\Phi : (x, G) \mapsto \phi_G(x) \in \mathbb{R}^h$. With this kernel, the empirical $\mathrm{MMD}_{k_f}^2$ between source and target node-feature pairs *reproduces* Eq. (1).

**GSS kernel consistent with Eq. (2).** For each node $i$ in graph $G$, the vector $[\mathbf{A}_G \cdot \mathbf{A}_G]_i$ has length $N_G$. To ensure well-defined comparisons when $N_o \neq N_t$, fix a *size-harmonization operator* $\rho : \mathbb{R}^{N_\bullet} \to \mathbb{R}^p$ (e.g., truncation/aggregation of walk-counts, heat signatures at $p$ scales, or any domain-agnostic deterministic map), applied identically across domains and independent of the data. Define

$$s_i^G := \rho\big([\mathbf{A}_G \cdot \mathbf{A}_G]_i\big) \in \mathbb{R}^p, \qquad k_g\big((i, G), (j, G')\big) := \exp\Big( - \tfrac{\| s_i^G - s_j^{G'} \|^2}{2\sigma^2} \Big).$$

When $N_o = N_t$ and $\rho$ is the identity, Eq. (2) is recovered verbatim; otherwise, Eq. (2) is interpreted with the implicit $\rho$ for well-definedness.

## A.3  Definitions of FSS and GSS (as MMDs)

Let $\mathcal{X}^o = \{(x_i^o, G_o)\}_{i=1}^{N_o}$ and $\mathcal{X}^t = \{(x_j^t, G_t)\}_{j=1}^{N_t}$ be source/target node-feature pairs; let $\mathcal{S}^o = \{(i, G_o)\}_{i=1}^{N_o}$ and $\mathcal{S}^t = \{(j, G_t)\}_{j=1}^{N_t}$ be the node-index pairs for structure. We define

$$\mathtt{FSS} := \mathrm{MMD}_{k_f}^2\big(\mathcal{X}^o, \mathcal{X}^t\big), \tag{A.2}$$

$$\mathtt{GSS} := \mathrm{MMD}_{k_g}^2\big(\mathcal{S}^o, \mathcal{S}^t\big). \tag{A.3}$$

By construction, (A.2) reproduces Eq. (1); (A.3) coincides with Eq. (2) when $N_o = N_t$ and $\rho$ is identity, and otherwise implements its size-harmonized counterpart.

## A.4 Additive Kernel and Decomposition

For the joint representation $U = ((x, G), (i, G))$ with feature part $\Phi(x, G) = \phi_G(x)$ and structure part $s_i^G = \rho([\mathbf{A}_G \mathbf{A}_G]_i)$, define the additive kernel

$$k(U, U') = k_f((x, G), (x', G')) + k_g((i, G), (j, G')). \tag{A.4}$$

Its RKHS is the orthogonal direct sum $\mathcal{H}_k = \mathcal{H}_{k_f} \oplus \mathcal{H}_{k_g}$, hence the mean embeddings decompose. Let $P_s^U, P_t^U$ be the joint distributions of $U$ on source/target. Then

$$\mathrm{MMD}_k^2(P_s^U, P_t^U) = \mathrm{MMD}_{k_f}^2(P_s^\Phi, P_t^\Phi) + \mathrm{MMD}_{k_g}^2(P_s^S, P_t^S) = \texttt{FSS} + \texttt{GSS}. \tag{A.5}$$

## A.5 Key Properties

**Theorem 1** (Non-negativity and identity of indiscernibles). *For positive-definite kernels $k_f, k_g$, $\texttt{FSS} \geq 0$ and $\texttt{GSS} \geq 0$. If $k_f$ (resp. $k_g$) is characteristic on its input space, then $\texttt{FSS} = 0$ (resp. $\texttt{GSS} = 0$) iff the corresponding source and target distributions coincide.*

*Proof.* Immediate from MMD as the RKHS distance between mean embeddings; characteristic kernels yield injective mean embeddings [45]. $\square$

## A.6 Domain-Adaptation Bound via MMD

We bound the generalization gap using the IPM property of MMD under explicit assumptions.

**Assumption 1 (Covariate shift).** $P_s(Y \mid U = u) = P_t(Y \mid U = u)$ for all $u$ in the support of $U$.

**Assumption 2 (RKHS capacity on conditional risk).** For each hypothesis $h$, define $g_h(u) = \mathbb{E}_{Y|U=u}[\ell(h(u), Y)]$ and assume $g_h \in \mathcal{H}_k$ with $\|g_h\|_{\mathcal{H}_k} \leq B$. Assume also that the loss is bounded, $0 \leq \ell \leq M$ (used when relating empirical and population risks).

**Theorem 2** (Generalization gap under additive-kernel discrepancy). *Under Assumptions 1–2,*

$$\left| \epsilon_t(h) - \epsilon_s(h) \right| \leq B \, \mathrm{MMD}_k(P_s^U, P_t^U) = B \sqrt{\texttt{FSS} + \texttt{GSS}}. \tag{A.6}$$

*Moreover, with probability at least $1 - \delta$ over i.i.d. draws of $N_o$ source and $N_t$ target samples,*

$$\epsilon_t(h) \leq \epsilon_s(h) + B \, \widehat{\mathrm{MMD}}_k(\mathcal{S}^U, \mathcal{T}^U) + C \sqrt{\frac{\ln(2/\delta)}{N_o + N_t}}, \tag{A.7}$$

*for a constant $C = C(B, \kappa)$ depending only on the RKHS radius $B$ and the kernel bound $\kappa$.*

*Proof sketch.* Under Assumption 1, $\epsilon_d(h) = \mathbb{E}_{U \sim P_d^U}[g_h(U)]$ for $d \in \{s, t\}$ with the same $g_h$. By the IPM characterization of MMD, $\sup_{\|f\|_{\mathcal{H}_k} \leq 1} |\mathbb{E}_s f(U) - \mathbb{E}_t f(U)| = \mathrm{MMD}_k(P_s^U, P_t^U)$. Since $\|g_h\|_{\mathcal{H}_k} \leq B$, we obtain (A.6); (A.7) follows by replacing the population MMD with its empirical counterpart and applying concentration for bounded kernels, yielding a deviation term that depends on $B$ and $\kappa$. $\square$

## A.7 Conclusion

This appendix formalized **FSS** and **GSS** as squared MMDs computed with well-defined kernels on shared, fixed-dimension spaces. Concretely, we: (1) specified a single positive-definite feature kernel $k_f$ induced by the shared encoder (parameters fixed when computing MMD), yielding a valid empirical $\mathrm{MMD}_{k_f}^2$; (2) introduced a size-harmonization operator to define a structure kernel $k_g$ consistently across graphs of different sizes; and (3) established the additive-kernel identity $\mathrm{MMD}_k^2 = \texttt{FSS} + \texttt{GSS}$ for $k = k_f + k_g$. Under covariate shift and an RKHS capacity assumption on the conditional risk, we derived the domain-adaptation bound $|\epsilon_t(h) - \epsilon_s(h)| \leq B \sqrt{\texttt{FSS} + \texttt{GSS}}$, with an empirical counterpart including a standard concentration term depending only on $B$ and $\kappa$. These results justify minimizing both FSS and GSS in our algorithm design—respectively via invariant feature learning and structure-insensitive affinity learning—to reduce the generalization gap on unseen graphs.

# B  Experimental Analysis of FSS and GSS

To further understand how IA-GGAD tackles domain shifts, we analyze the quantitative behaviors of FSS and GSS as defined in Eq. (1) and Eq. (2), and examine the effect of the final score fusion strategy defined in Eq. (13). As shown in Fig. 4, our anomaly scoring mechanism effectively mitigates the domain shift challenges posed by FSS and GSS. By adaptively weighting the residual-based semantic score and the affinity-based structural score, IA-GGAD balances two orthogonal cues, resulting in robust performance across both FSS-dominated and GSS-dominated datasets.

## B.1  FSS Scores Analysis

Table 5: FSS scores from source domains to target domains.

| Source \ Target | ACM | BlogCatalog | Reddit | Facebook | Weibo | Cora | Amazon | Citeseer |
|---|---|---|---|---|---|---|---|---|
| **Flickr** | 0.0153 | 0.0100 | 0.0018 | 0.3616 | 0.5624 | 0.5082 | 0.0295 | 0.7178 |
| **YelpChi** | 0.0134 | 0.0124 | 0.0087 | 0.3601 | 0.5608 | 0.5067 | 0.0253 | 0.7163 |
| **Pubmed** | 0.0074 | 0.0137 | 0.0140 | 0.3529 | 0.5521 | 0.4975 | 0.0285 | 0.7072 |
| **Questions** | 0.0494 | 0.0523 | 0.0526 | 0.3231 | 0.5175 | 0.4623 | 0.0452 | 0.6716 |
| **Mean** | 0.0214 | 0.0221 | 0.0193 | 0.3494 | 0.5482 | 0.4937 | 0.0321 | 0.7032 |

**Feature Space Shift (FSS).**  Table 5 reports the average FSS scores between each source and target domain, providing a quantitative measure of the distributional misalignment in the node feature space. Based on the average FSS values, we categorize the target datasets into three types of FSS domains:

- **Low-FSS domains (FSS < 0.05)**: **Reddit** (0.0193), **ACM** (0.0214), **BlogCatalog** (0.0221), and **Amazon** (0.0321) exhibit minimal feature space shift from the source domains. Their node attributes are highly compatible with those in the training graphs, likely due to shared semantics such as user interactions, co-occurrence structures, or platform-generated tags. As such, these domains allow for direct knowledge transfer with negligible adaptation cost.

- **Moderate-FSS domains ($0.05 \leq$ FSS < 0.4)**: **Facebook** (0.3494) demonstrates moderate misalignment in its node feature space. While not as challenging as high-FSS domains, the shift indicates partial semantic divergence, potentially stemming from demographic-specific behaviors or inconsistent attribute ontologies. Alignment strategies are still necessary to ensure effective transfer.

- **High-FSS domains (FSS $\geq$ 0.4)**: **Cora** (0.4937), **Weibo** (0.5482), and **Citeseer** (0.7032) represent severely misaligned domains with pronounced semantic drift. Such high FSS scores suggest substantial differences in feature distributions, likely caused by sparse vocabulary, heterogeneous encodings, or conflicting representation schemes (e.g., bag-of-words vs. contextual embeddings). These domains demand robust invariant encoding mechanisms to support generalization under extreme domain shifts.

## B.2  GSS Scores Analysis

Table 6: GSS scores from source domains to target domains.

| Source \ Target | ACM | BlogCatalog | Reddit | Facebook | Weibo | Cora | Amazon | Citeseer |
|---|---|---|---|---|---|---|---|---|
| **Flickr** | 0.1128 | 0.0372 | 0.1929 | 0.0123 | 0.1812 | 0.3711 | 0.0837 | 0.6624 |
| **YelpChi** | 0.5016 | 0.5316 | 0.3651 | 0.4404 | 0.4587 | 0.1254 | 0.1938 | 0.0270 |
| **Pubmed** | 0.7689 | 0.7173 | 0.6165 | 0.6375 | 0.6327 | 0.3162 | 0.3750 | 0.0360 |
| **Questions** | 0.9042 | 0.7938 | 0.8238 | 0.7431 | 0.3258 | 0.6303 | 0.5856 | 0.4626 |
| **Mean** | 0.5719 | 0.5200 | 0.4996 | 0.4583 | 0.3996 | 0.3608 | 0.3095 | 0.2970 |

**Graph Structure Shift (GSS).**    Table 6 reports the average GSS scores from each source domain to each target domain, quantifying the degree of structural distributional shift. Based on these scores, we classify the target domains into three levels of structural misalignment:

- **Low-GSS domains (GSS $< 0.35$)**: **Amazon** (0.3095) and **Citeseer** (0.2970) exhibit minimal structural deviation from the source graphs. Their graph topologies—such as degree distributions, community structures, and connectivity statistics—align closely with those seen in training domains. These domains are structurally compatible and require little to no adaptation for generalization.

- **Moderate-GSS domains ($0.35 \leq$ GSS $< 0.5$)**: **Cora** (0.3608), **Weibo** (0.3996), **Facebook** (0.4583), and **Reddit** (0.4996) fall into the moderate shift category. These domains show partial topological divergence, possibly due to differences in local density, edge formation policies, or subgraph structures. Moderate adaptation via structure-aware encoders remains beneficial here.

- **High-GSS domains (GSS $\geq 0.5$)**: **BlogCatalog** (0.5200) and **ACM** (0.5719) exhibit strong structural misalignment. These domains likely differ in both macro-scale topology (e.g., degree skewness, small-worldness) and micro-scale motifs. As a result, traditional structural priors become unreliable, necessitating robust affinity modeling or structure-invariant mechanisms for effective transfer.

### B.3    Empirical Evidence

**The Solution of FSS and GSS Challenges.**    To balance feature space shift and graph structure shift across domains, we adopt a weighted fusion scheme (Eq. 13) controlled by a weighting factor $\lambda \in [0, 1]$. Our empirical study (Fig. 4) reveals a clear correspondence between the optimal choice of $\lambda$ and the underlying FSS/GSS characteristics of each dataset. Specifically, datasets suffering from substantial semantic shift, such as **Citeseer** (FSS = 0.7032, GSS = 0.2970) and **Cora** (FSS = 0.4937, GSS = 0.3608)—achieve peak performance at lower values of $\lambda$ (e.g., 0.1–0.3). This indicates that the residual-based semantic score $\mathcal{RS}(v_i)$ plays a dominant role in these scenarios, where invariant feature alignment is crucial for mitigating cross-domain semantic discrepancies. In contrast, structurally misaligned domains like **ACM** (FSS = 0.0214, GSS = 0.5719) and **Facebook** (FSS = 0.3494, GSS = 0.4583) require larger $\lambda$ values (e.g., 0.7–0.9), reflecting a stronger dependence on the affinity-based structural score $\mathcal{AS}(v_i)$. In these cases, topological deviations from the source domains dominate, making structural modeling the primary means of anomaly detection. These findings confirm that our fusion mechanism flexibly adapts to the dominant domain shift type—semantic or structural—thereby enabling robust zero-shot generalization without manual tuning.

## C    Algorithm and Complexity

### C.1    Algorithmic description

The algorithmic description of the training and inference process of IA-GGAD is summarized in Algorithm. 1, and Algorithm. 2, respectively.

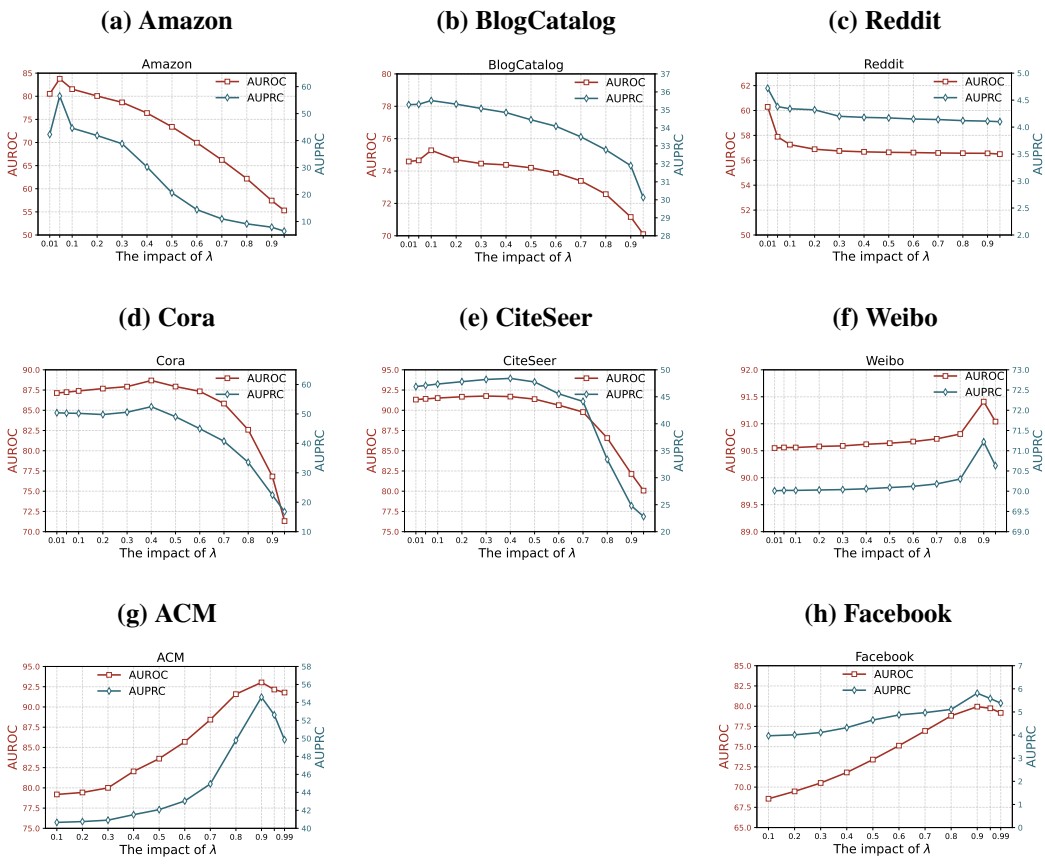

Figure 4: Impact of anomaly score weighting factor $\lambda$ on AUROC and AUPRC across different datasets.

---

**Algorithm 1:** The Training algorithm of IA-GGAD

---

**Input:** Training datasets $\mathcal{T}_{train}$.

**Parameters :** Number of epoch $T$; GCNs layers: $\ell$ .

1   Generate initial parameters for all learnable parameters.

2   **for** $\mathcal{D} \in \mathcal{T}_{train}$ **do**

3     Obtain aligned features $\bar{\mathbf{X}}$ from $\mathbf{X}$ via feature alignment.

4   **end**

5   **for** each epoch $t = 0, 1, 2, ..., T$ **do**

6     **for** $\mathcal{D} \in \mathcal{T}_{train}$ **do**

7       Obtain $\bar{\mathbf{X}}, \mathbf{A}, \mathcal{V}, \mathbf{y}$ from $\mathcal{D}$

8       $\mathbf{Z}^{[1]}, \cdots, \mathbf{Z}^{[\ell]} \leftarrow$ GCNs transform embedding via Eq. (3)

9       Initial and update $E = \{e_1, e_2, \cdots, e_M\}$ with $\mathbf{Z}^{[l]}$ via Eq. (4)

10      **for** $\ell = 1 : L$ **do**

11        $h_i^{[\ell]} \leftarrow$ concat $\mathbf{Z}^{[l]}$ and $E$ via Eq. (6)

12        $r_i \leftarrow$ Calculate residual of $h_i^\ell$ and $h_i^{[1]}$ via Eq. (7)

13      **end**

14      Calculate loss $\mathcal{L}_{inv}$ and $\mathcal{L}_{res}$ via Eq. (5) and Eq. (8)

15      Obatain node affinity emebding $\bar{\mathbf{H}}$ via Eq. (9)

16      Calculate nodes affinity score $\mathcal{AS}(v_i)$ by Eq. (10)

17      Maximize node affinity by $\mathcal{L}_{aff}$ Eq.(11)

18      Update model parameters via gradient descent.

19     **end**

20   **end**

---

**Algorithm 2:** The Inference algorithm of IA-GGAD

---

**Input:** Test dataset $\mathcal{T}_{test}$
**Parameters:** Well-trained IA-GGAD model weight parameters.

1 **for** $\mathcal{D} \in \mathcal{T}_{test}$ **do**
2     Obtain aligned features $\bar{\mathbf{X}}$ from $\mathbf{X}$ via feature alignment.
3 **end**
4 Obtain $\bar{\mathbf{X}}, \mathbf{A}, \mathcal{V}$ from $\mathcal{D}$
5 $\mathbf{Z}^{[1]}, \cdots, \mathbf{Z}^{[\ell]} \leftarrow$ GCNs transform embedding via Eq. (3)
6 **for** $l = 1 : L$ **do**
7     $h_i^{[\ell]} \leftarrow$ concat $\mathbf{Z}^{[l]}$ and pretrianed $E$ via Eq. (6)
8     $r_i \leftarrow$ Calculate residual of $h_i^\ell$ and $h_i^{[1]}$ via Eq. (7)
9 **end**
10 Obatain node affinity emebding $\bar{\mathbf{H}}$ via Eq. (9)
11 Calculate residual dispersion score $\mathcal{RS}(v_i)$ by Eq. (12)
12 Calculate nodes affinity score $\mathcal{AS}(v_i)$ by Eq. (10)
13 Joint anomaly scoring and prediction $\mathcal{S}(v_i)$ by Eq. (13)

---

### C.2 Complexity Analysis

**Training Phase.** The total time complexity in the training phase of IA-GGAD consists of four main components:

- **Feature alignment:** projecting raw node features $X \in \mathbb{R}^{n \times d}$ into a unified latent space with dimension $d_u$ via linear projection (e.g., PCA), resulting in complexity $\mathcal{O}(ndd_u)$.

- **Graph representation learning:** using $L$-layer GCNs for message passing and residual encoding. This includes feature propagation $\mathcal{O}(md_u)$ and residual encoding $\mathcal{O}(nd_uh + nh^2)$, leading to an overall complexity of $\mathcal{O}(L(md_u + nd_uh + nh^2))$.

- **Invariant prototype updating:** constructing and updating a invariant features pool $E$ of $M$ domain-invariant features using nearest neighbor assignment: $\mathcal{O}(nMd_u)$.

- **Affinity learning:** calculating affinity scores between each node and its neighbors using cosine similarity: $\mathcal{O}(n\bar{d}\,\bar{h})$, where $\bar{d}$ is the average node degree, $\bar{h}$ is the affinity emebding.

Thus, the total training complexity is:
$$\mathcal{O}\left(ndd_u + L(md_u + nd_uh + nh^2) + nMd_u + n\bar{d}\,\bar{h}\right)$$

In our experiments, the total training time across all datasets was approximately 9 minutes for five independent runs (with different random seeds), conducted on an NVIDIA GeForce RTX A6000 GPU.

**Testing Phase.** The inference process involves feature alignment, embedding generation, and anomaly scoring:

- **Feature alignment:** similar to training, $\mathcal{O}(ndd_u)$ for projection, smoothness computation, and feature reordering.

- **Embedding generation:** $\mathcal{O}(L(md_u + nd_uh + nh^2))$, as in training.

- **Anomaly scoring:**
  - Residual dispersion scoring via $k$ sample in residual space: $\mathcal{O}(nkh)$.
  - Affinity scoring via neighbor cosine similarity: $\mathcal{O}(n\bar{d}h)$.
  - Weighted fusion and thresholding: $\mathcal{O}(n)$.

Therefore, the total inference complexity is:
$$\mathcal{O}\left(ndd_u + L(md_u + nd_uh + nh^2) + nkh + n\bar{d}\,\bar{h}\right)$$

In our experiments, the total inference time across all datasets was approximately 3–5 seconds when performed on an NVIDIA GeForce RTX A6000 GPU.

# D   Detailing Related Work

**Anomaly Detection.** Anomaly detection (AD) aims to identify samples that deviate significantly from the norm [5]. Due to the scarcity of labeled anomalies, most AD methods operate in an unsupervised manner. To extract anomaly patterns without supervision, existing approaches leverage techniques such as one-class classification [46], distance-based scoring [47], reconstruction [15], generative modeling [48], and self-supervised learning [16]. For instance, DeepSVDD [46] formulates an objective to learn compact representations within a hypersphere, capturing the common modes of normal data. AnoDDPM [48] adopts a partial diffusion process with simplex noise to detect anomalies effectively, particularly in high-resolution image settings. While effective, these methods are typically specialized for the domain on which they are trained, limiting their ability to generalize to unseen datasets or cross-domain scenarios.

**Cross-Dataset Anomaly Detection.** To overcome dataset-specific limitations, recent anomaly detection (AD) research has explored generalization across domains. Some methods tackle domain shifts by leveraging distribution alignment or domain adaptation [49, 50], yet they often assume semantic similarity between source and target domains, which limits broader applicability. A more flexible strategy is the few-shot setting, where a small number of normal samples from the target domain are available to guide detection. Under this paradigm, RegAD [51] learns a transferable model that generalizes to new domains without retraining. WinCLIP [52] and InCTRL [53] exploit vision-language models (VLMs) such as CLIP to achieve zero- or few-shot image anomaly detection. Despite their impressive performance on image data, these methods rely heavily on pre-trained vision encoders and tailored architectures, limiting their transferability to graph-based domains.

**Anomaly Detection on Graph Data.** Depending on the granularity of anomalies, graph-based anomaly detection (AD) methods can be broadly classified into three categories: node-level [15, 3], edge-level [54], and graph-level [55, 56]. Among these, node-level AD has attracted the most attention due to its wide applicability in real-world scenarios [6]. In this work, we focus specifically on node-level anomaly detection and adopt the term Graph Anomaly Detection (GAD), in line with common usage in prior literature [29, 3, 23].

**Graph Anomaly Detection.** Early graph anomaly detection (GAD) methods primarily relied on shallow techniques. For instance, AMEN [17] detects anomalies by modeling attribute correlations within each node's ego-network. Residual analysis was also commonly employed, such as in Radar [18], which evaluates inconsistencies between node attributes and structural information. ANOMALOUS [19] further integrates attribute selection with anomaly scoring via CUR decomposition. Although these methods achieved reasonable performance on low-dimensional graphs, they struggle with complex structures and high-dimensional attributes due to their limited representational capacity [57, 58]. To overcome the limitations of shallow methods, graph neural networks (GNNs) have become the dominant paradigm for graph anomaly detection (GAD). Existing GNN-based approaches can be broadly categorized into supervised and unsupervised settings [6, 35]. Supervised GAD assumes access to labeled normal and anomalous nodes, and research in this direction primarily focuses on improving convolutional architectures and task-specific objective functions [59, 3]. For example, CARE-GNN [59] enhances fraud detection by incorporating label-aware neighbor aggregation via reinforcement learning to combat disguise attacks. Spectral methods provide another perspective, linking anomalies to high-frequency signals in the graph spectrum. BWGNN [3] introduces band-pass filters to capture localized spectral patterns, while GHRN [26] further emphasizes high-frequency components by pruning inter-class edges, isolating anomalous nodes more effectively. Unlike supervised methods, unsupervised graph anomaly detection (GAD) does not rely on labeled data. Inspired by unsupervised anomaly detection in images, these approaches adopt various learning paradigms—such as reconstruction, contrastive learning, and auxiliary pretext tasks—to uncover node-level anomalies. For example, DOMINANT [15] leverages a graph autoencoder to jointly reconstruct adjacency and attribute matrices, detecting anomalies via reconstruction errors. ComGA [23] enhances detection by integrating community structure and tailored GCNs to capture local and structural anomalies. CoLA [16] introduces contrastive self-supervised learning into GAD, constructing instance pairs without labels to learn discriminative node representations. HCM-A [37] incorporates hop prediction and Bayesian learning to integrate multiscale context. More recently, TAM [29] proposes a homophily-aware affinity score, optimized end-to-end on a truncated graph structure to better isolate anomalous nodes. GCTAM[30] extends one-class deep learning to graphs by optimizing an affinity objective over global GNN embeddings.

**Generalist Graph Anomaly Detection.** Nevertheless, all the above methods adhere to the conventional paradigm of "one-for-one" dataset. Recent work proposes generalist graph anomaly detection frameworks. ARC [20], a "one-for-all" GAD framework based on in-context learning, which aligns node features from different graphs using a learned feature-space projection, encodes residual neighborhood patterns via an ego-neighbor graph encoder, and employs a cross-attentive scoring module that compares nodes to a few-shot set of normal examples. UNPrompt [22], a zero-shot generalist GAD model trained on a single source graph, which aligns the dimensionality and semantics of node attributes across graphs through coordinate-wise normalization and learns generalized neighborhood prompts so that the predictability of latent node attributes serves as a universal anomaly score. Despite their contributions, ARC and UNPrompt have limitations. ARC still requires a few target-domain normal examples at inference, and its learned alignment may not eliminate all domain shifts. UNPrompt relies on consistent attribute semantics and a single-source training setup, which can limit its applicability when graphs vary widely in feature space or structure. In contrast, our IA-GGAD directly addresses these gaps. IA-GGAD is a zero-shot generalist GAD framework requiring no target-specific data or fine-tuning.

## E  Description of Datasets

Following ARC [20], we evaluate our model on 12 benchmark datasets, categorized into four groups: **(1)** citation networks with injected anomalies, **(2)** social networks with injected anomalies, **(3)** social networks with real anomalies, and **(4)** co-review networks with real anomalies. For each category, we designate the largest dataset as the training source, while the remaining datasets serve as testing targets. This setting enables a comprehensive evaluation of the generalization capability of our proposed IA-GGAD model.

Table 7 summarizes the statistics of all datasets. The selected datasets span diverse domains and include both synthetic and real-world anomalies, ensuring the model is exposed to a wide range of anomaly types. This diversity is essential for equipping IA-GGAD with the ability to generalize effectively to unseen graphs. Detailed descriptions of each dataset are provided below.

Table 7: The statistics of datasets.

| Dataset | Train | Test | #Nodes | #Edges | #Features | Avg. Degree | #Anomaly | %Anomaly |
|---------|-------|------|--------|--------|-----------|-------------|----------|----------|
| Citation network with injected anomalies | | | | | | | | |
| Cora | - | ✓ | 2,708 | 5,429 | 1,433 | 3.90 | 150 | 5.53 |
| CiteSeer | - | ✓ | 3,327 | 4,732 | 3,703 | 2.77 | 150 | 4.50 |
| ACM | - | ✓ | 16,484 | 71,980 | 8,337 | 8.73 | 597 | 3.62 |
| PubMed | ✓ | - | 19,717 | 44,338 | 500 | 4.50 | 600 | 3.04 |
| Social network with injected anomalies | | | | | | | | |
| BlogCatalog | - | ✓ | 5,196 | 171,743 | 8,189 | 66.11 | 300 | 5.77 |
| Flickr | ✓ | - | 7,575 | 239,738 | 12,047 | 63.30 | 450 | 5.94 |
| Social network with real anomalies | | | | | | | | |
| Facebook | - | ✓ | 1,081 | 55,104 | 576 | 50.97 | 25 | 2.31 |
| Weibo | - | ✓ | 8,405 | 407,963 | 400 | 48.53 | 868 | 10.30 |
| Reddit | - | ✓ | 10,984 | 168,016 | 64 | 15.30 | 366 | 3.33 |
| Questions | ✓ | - | 48,921 | 153,540 | 301 | 3.13 | 1,460 | 2.98 |
| Co-review network with real anomalies | | | | | | | | |
| Amazon | - | ✓ | 10,244 | 175,608 | 25 | 17.18 | 693 | 6.76 |
| YelpChi | ✓ | - | 23,831 | 49,315 | 32 | 2.07 | 1,217 | 5.10 |

- **Cora, CiteSeer, PubMed**[60] and **ACM**[61] are four widely-used citation network datasets. In these datasets, nodes correspond to scientific publications, and edges represent citation relationships between them. Each node is described by a bag-of-words feature vector, where the dimensionality is determined by the size of the vocabulary specific to each dataset.

- **BlogCatalog** and **Flickr** [15] are representative social network datasets, where users are connected via mutual following relationships. Each user is represented as a node, and edges denote social connections. Node attributes are derived from user-generated textual content within the platform, including blog posts, photo tags, and other descriptive metadata.

- **Amazon** and **YelpChi**[62, 63] are datasets that capture user-review interactions to identify opinion fraud. The Amazon dataset is constructed to detect users who were incentivized to post fake product reviews. Following prior work[29], three graph variants are derived from Amazon using different relational schemes to form adjacency matrices. YelpChi, on the other hand, focuses on detecting deceptive reviews on Yelp.com that unfairly promote or defame businesses. Based on [62, 64], three graph variants are also constructed for YelpChi, incorporating relationships among users, review content, and timestamps. In this study, we specifically adopt the Amazon-UPU variant (where edges connect users who reviewed at least one common product) and the YelpChi-RUR variant (where edges connect reviews posted by the same user).

- **Facebook [65]** is a social network in which users can build relationships with others and share their friends.

- **Reddit** [66] is a forum-based social network dataset collected from the Reddit platform. In this dataset, users who have been banned are labeled as anomalies. Each node represents a user, and edges reflect interactions such as replies or shared threads. The textual content of user posts is encoded into vector representations and used as node attributes.

- **Weibo** [66] is a social media dataset derived from the Tencent Weibo platform, comprising a graph of users and their associated hashtags. Within a defined temporal window (e.g., 60 seconds), if a user posts consecutively, the behavior is considered potentially suspicious. Users who exhibit at least five such instances are labeled as "suspicious" and treated as anomalies. Node features include geolocation data of microblog posts and bag-of-words representations of the textual content.

- **Questions** [67] dataset originates from Yandex Q, a platform dedicated to question-answering. Users represent the nodes, while the connections between them signify the presence or absence of a question-and-answer interaction within a one-year timeframe. Node features are constructed by averaging the FastText embeddings of the words in each user's profile description. An additional binary feature is included to denote users with missing descriptions.

**Anomaly Injection.** For datasets with injected anomalies, we follow the injection strategy introduced in [15, 16]. Specifically, we directly adopt the publicly available datasets from ARC [20], in which anomalies have already been injected using standardized procedures. In summary, the injection process perturbs both graph structure and node attributes. Structurally, anomalous cliques are created by densely connecting randomly selected nodes, simulating unnatural substructures. For attribute perturbations, features of selected nodes are replaced with those from the most dissimilar nodes to ensure a significant semantic shift. The total number of anomalies is controlled proportionally to the dataset size. More detailed statistics of all datasets are summarized in Table 7.

# F    Description of Baselines

In our evaluation, we present a comprehensive comparison of IA-GGAD against a variety of graph anomaly detection (GAD) methods, including supervised, semi-supervised, and unsupervised paradigms. We also include comparisons with recent SOTA generalist GAD (GGAD) approaches, which aim to perform anomaly detection across diverse datasets using a single unified model.

**Supervised Method.** For the supervised setting, we consider two classical GNN architectures as well as four state-of-the-art (SOTA) models specifically developed for the GAD task. These methods assume access to labels for both normal and anomalous nodes during training. Accordingly, the problem is framed as a binary node classification task, where the goal is to accurately distinguish anomalous nodes from normal ones.

- **GCN** [14] is a seminal model in the development of graph neural networks (GNNs). It leverages neighborhood aggregation to effectively capture local graph structure, enabling efficient node feature extraction and representation learning for graph-structured data.

- **GAT** [36] introduces an attention mechanism into the GNN framework, allowing the model to dynamically assign weights to neighboring nodes. This enhances its adaptability across downstream tasks by producing context-aware node representations.

- **BGNN** [25] integrates gradient boosted decision trees (GBDTs) with GNNs to effectively handle graphs with tabular node features. While GBDTs manage feature heterogeneity, the GNN component captures structural dependencies, leading to superior performance on mixed-type data.

- **BWGNN** [3] utilizes spectrally and spatially localized band-pass filters to address the "right-shift" phenomenon in graph anomalies, where abnormal nodes tend to exhibit high-frequency spectral energy concentrations.

- **GHRN** [26] is a heterophily-aware supervised GAD model based on spectral analysis. By enhancing high-frequency signals and pruning inter-class edges, GHRN effectively isolates anomalous nodes and improves detection performance.

- **CAGAD** [27] employs a graph pointer network to identify heterophilic anomalies—nodes embedded in neighborhoods dominated by normal nodes. It generates counterfactual representations by aggregating information from unseen neighbors, enhancing anomaly detection in an unsupervised manner.

**Semi-supervised Method.** For the semi-supervised method, S-GAD [28], a recently proposed method specifically designed for graph anomaly detection with access to only a small subset of labeled normal nodes.

- **S-GAD** [68] is a semi-supervised GAD method that uses a few labeled normal nodes to train a one-class classifier. It generates learnable pseudo-anomalies based on asymmetric local affinity and egocentric closeness, which serve as negative samples to enhance anomaly detection without requiring labeled anomalies.

**Unsupervised Methods.** For the unsupervised methods, we consider 5 representative SOTA GAD methods, each of them belonging to a sub-type: data reconstruction, contrastive learning, hop-based auxiliary goal, or affinity-based auxiliary goal:

- **DOMINANT** [15] combines GCN and deep auto-encoder, and its learning objective is to reconstruct the adjacency matrix and node features jointly. It aims to identify structural and attribute anomalies based on reconstruction errors.

- **CoLA** [16] is a contrastive self-supervised learning for anomaly detection on graphs with node attributes. The framework captures the relationship between each node and its neighborhood substructure in an unsupervised manner by sampling novel pairs of contrasting instances and leveraging the local information of the graph.

- **HCM-A** [37] uses hop-count prediction as a self-supervised task to better identify anomalies by modeling both local and global context information. In addition, HCM-A designs two new anomaly scores and introduces Bayesian learning to train the model to capture anomalies.

- [1]**TAM** [29] is designed based on one-class homophily and local affinity. The learning target of TAM is to optimize the proposed anomaly metric (i.e., affinity) end-to-end on the truncated adjacency matrix.

- [2]**GCTAM** [30] is an unsupervised GAD method that enhances truncated affinity maximization by combining contextual and global affinity truncation. It introduces two key modules: contextual affinity truncation (CAT), which reduces the influence of anomalous nodes by cutting weak contextual links, and global affinity truncation (GAT), which enhances affinity among normal nodes. By integrating both modules through shared GCNs, GCTAM generates node representations that better reflect homophily and irregularity, significantly boosting anomaly detection performance across real-world datasets.

**Generalist GAD methods.** For the generalist setting, we consider 2 state-of-the-art GGAD methods that aim to detect anomalies across diverse domains using a unified model. These methods do not rely on dataset-specific training or adaptation and are designed to generalize to unseen graphs via in-context learning, prototype alignment:

---

[1]TAM: `https://github.com/mala-lab/TAM-master`

[2]GCTAM: `https://github.com/kgccc/GCTAM`

[3]ARC: `https://github.com/yixinliu233/ARC`

[4]UNPrompt: `https://github.com/mala-lab/UNPrompt`

- [3]**ARC** [20] is a few-shot generalist GAD method based on in-context learning. It uses a residual graph encoder to extract anomaly-aware node features and a cross-attention module to reconstruct query nodes from a few labeled normal context nodes. Anomaly scores are computed by measuring residual distance between original and reconstructed embeddings, allowing ARC to detect anomalies across unseen graphs without fine-tuning.

- [4]**UNPrompt** [22] is a zero-shot generalist GAD method that unifies node attributes across graphs via coordinate-wise normalization and learns transferable normal/abnormal patterns through neighborhood prompt learning. It performs anomaly detection by measuring latent attribute predictability without any training or labels on the target graphs.

# G  Details of Implementation

**Hyper-parameters.** We select some key hyperparameters of IA-GGAD through random search within specified grids. Specifically, the random search was performed within the following search space:

- Hidden layer dimension: $\{64, 128, 256, 512, 1024\}$
- Number of invariant and affinity encoder layers: $\{1, 2, 3\}$
- Dropout rate: $\{0, 0.1, 0.2, 0.3, 0.4, 0.5, 0.6, 0.7, 0.8\}$
- Learning rate: floats between $10^{-5}$ and $10^{-2}$
- Weight decay: floats between $10^{-6}$ and $10^{-3}$
- Number of invariant features: $\{1024, 2048, 4096, 8192\}$

**Baseline Implementation.** We adopt a unified set of hyperparameters to construct a generalist GAD model applicable across all datasets. All methods, including the proposed IA-GGAD and baseline models, are first trained on the source training set $\mathcal{T}$train using full anomaly labels. Subsequently, each model is evaluated independently on every dataset in the target test datasets $\mathcal{T}$test, without any retraining or fine-tuning. For feature projection, we apply Principal Component Analysis (PCA) to map the raw node attributes into a fixed-dimensional latent space with $d_u = 64$. In cases where the original feature dimension is smaller than $d_u$, we first apply a random projection (e.g., Gaussian random projection) to upscale the features, followed by PCA to ensure uniform dimensionality alignment at $d_u$. For CAGAD[27], S-GAD[28], GCTAM [30], and UNPrompt [22], we reproduce their results using their official implementations and conduct optimal hyperparameter tuning. For all other baselines, we follow the reproduction settings reported in ARC [20]. It is worth noting that all methods are trained and evaluated under the same standardized experimental pipeline to ensure fair comparison.

**Metrics.** Following [35, 29, 38], we employ two popular and complementary evaluation metrics for evaluation, including area under the receiver operating characteristic Curve (AUROC) and area under the precision-recall curve (AUPRC). A higher AUROC/AUPRC value indicates better performance. We report the average AUROC/AUPRC with standard deviations across 5 runs.

**Implementation Details.** The experiments in this study were conducted on a Linux server running Ubuntu 20.04. The server was equipped with a 13th Gen Intel(R) Core(TM) i7-12700 CPU, 64GB of RAM, and an NVIDIA GeForce RTX A6000 GPU (48GB memory). For software, we used Anaconda3 to manage the Python environment and PyCharm as the development IDE. The specific software versions were Python 3.8.14, CUDA 11.7, DGL 0.9.1, and PyTorch 2.0.1 [69].

# H  Supplementary Experiments

## H.1  Performance Comparison of AUPRC

Table 8 presents the anomaly detection performance in terms of AUPRC across eight target datasets. IA-GGAD achieves the **best overall ranking (1.87)**, consistently outperforming both conventional GAD baselines and recent generalist methods. It ranks **first on five** datasets (ACM, Amazon, Cora, CiteSeer, Weibo) and **second on two** (BlogCatalog, Reddit), demonstrating strong adaptability to diverse semantic and structural shifts. Notable gains over the strongest baseline, ARC, include

**+17.83%** on ACM, **+12.29%** on Amazon, and **+6.77%** on Weibo. On Facebook—ARC's strongest domain—IA-GGAD remains competitive (6.55% vs. 8.38%, **–1.83%** gap). While ARC ranks second overall, its performance varies across datasets. GCTAM performs well on ACM and Facebook but fails to generalize under structural shift. Traditional models like GCN, GAT, and BGNN perform poorly, particularly on complex graphs such as Reddit and Weibo, highlighting their limited transferability. These results validate the effectiveness of IA-GGAD's fusion of invariant semantic features and structure-aware affinity encoding, enabling robust zero-shot anomaly detection across heterogeneous domains.

Table 8: Anomaly detection performance in terms of AUPRC (in percent, mean±std). Highlighted are the results ranked first, second, and third. "Rank" indicates the average ranking over 8 datasets. Methods with * representes reproduce results, others are report results in ARC[20].

| Method | ACM | Facebook | Amazon | Cora | CiteSeer | BlogCatalog | Reddit | Weibo | Rank |
|---|---|---|---|---|---|---|---|---|---|
| **GAD methods** | | | | | | | | | |
| GCN(2017) | 5.27±1.12 | 1.59±0.11 | 6.96±2.04 | 7.41±1.55 | 6.40±1.40 | 7.44±1.07 | 3.39±0.39 | 67.21±15.20 | 10.25 |
| GAT(2018) | 4.70±0.75 | 3.14±0.37 | 15.74±17.85 | 6.49±0.84 | 5.58±0.62 | 12.81±2.08 | 3.73±0.54 | 33.34±9.80 | 8.0 |
| BGNN(2021) | 3.48±1.33 | 3.81±2.12 | 7.51±0.58 | 4.90±1.27 | 3.91±1.01 | 5.73±1.47 | 3.52±0.50 | 30.26±29.98 | 11.25 |
| BWGNN(2022) | 7.14±0.20 | 2.54±0.63 | 13.12±11.82 | 7.25±0.80 | 6.35±0.73 | 8.99±1.12 | 3.69±0.81 | 12.13±0.71 | 9.75 |
| GHRN(2023) | 5.61±0.71 | 2.41±0.62 | 7.54±2.01 | 9.56±2.40 | 7.79±2.01 | 10.94±2.56 | 3.24±0.33 | 28.53±7.38 | 9.37 |
| DOMINANT(2019) | 15.59±2.69 | 2.95±0.06 | 6.11±0.29 | 12.75±0.71 | 13.85±2.34 | 35.22±0.87 | 3.49±0.44 | 81.47±0.22 | 6.0 |
| CoLA(2021) | 7.31±1.45 | 1.90±0.68 | 11.06±4.45 | 11.41±3.51 | 8.33±3.73 | 6.04±0.56 | 3.71±0.67 | 7.59±3.26 | 9.62 |
| HCM-A(2022) | 4.01±0.61 | 2.08±0.60 | 5.87±0.07 | 5.78±0.76 | 4.18±0.75 | 6.89±0.34 | 3.18±0.23 | 21.91±11.78 | 12.62 |
| TAM(2023) | 23.20±2.36 | 8.40±0.97 | 10.75±3.10 | 11.18±0.75 | 11.55±0.44 | 10.57±1.17 | 3.94±0.13 | 16.46±0.09 | 6.37 |
| CAGAD(2024)* | 7.97±4.67 | 2.61±0.76 | 3.49±0.73 | 5.31±3.20 | 3.85±1.60 | 6.40±3.06 | 13.56±18.91 | 20.95±18.34 | 10.37 |
| S-GAD(2024)* | 9.72±2.18 | 3.44±1.05 | 7.37±0.98 | 4.65±0.50 | 3.72±0.30 | 22.95±6.54 | 4.25±0.21 | 52.42±4.13 | 8.87 |
| GCTAM(2025)* | 48.09±0.28 | 9.61±0.96 | 13.71±0.11 | 9.52±0.69 | 10.29±0.28 | 27.47±0.54 | 4.34±0.17 | 16.87±1.31 | 5.0 |
| **GGAD methods** | | | | | | | | | |
| UNPrompt(2024)* | 10.45±1.55 | 2.61±0.45 | 10.27±7.04 | 6.02±0.2 | 4.47±0.32 | 24.89±3.25 | 5.15±0.65 | 18.67±4.33 | 7.87 |
| ARC(2024) | 40.62±0.10 | 8.38±2.39 | 44.25±7.41 | 49.33±1.64 | 45.77±1.25 | 36.06±0.18 | 4.48±0.28 | 64.18±0.55 | 2.62 |
| **IA-GGAD (ours)** | **58.45±1.32** | 6.55±0.41 | **56.54±10.50** | **52.47±0.83** | **48.71±1.39** | 35.52±0.19 | 4.74±0.48 | 70.95±0.30 | **1.87** |

