# OpenReview forum: "IA-GGAD: Zero-shot Generalist Graph Anomaly Detection via Invariant and Affinity Learning"
_NeurIPS.cc/2025/Conference — NeurIPS 2025 spotlight_

### Official Review · Reviewer_5Jb1 · 2025-06-05

**Clarity:** 2
**Significance:** 2
**Originality:** 1
**Rating:** 4
**Confidence:** 4

**Summary:**

By tackling Feature Space Shift and Graph Structure Shift, this paper proposes a unified framework, IA-GGAD, enabling anomaly prediction on unseen graphs without target-domain retraining or fine-tuning. Experiments show the improvements on given datasets to some extend.

**Questions:**

Q1. Can this framework be applied to larger datasets, such as T-Social and DGraph-Fin?

Q2. Can this framework be applied to cross-domain data? For example, will the performance drop a lot if it is trained on financial dataset and tested on social network? or reversely?

**Ethical Concerns:**

["NO or VERY MINOR ethics concerns only"]

**Final Justification:**

Although I still have concerns about the novelty, I have decided to increase my score to thank the authors.

**Limitations:**

Yes

**Quality:**

2

**Strengths And Weaknesses:**

S1. This paper proposes a unified framework, IA-GGAD, enabling anomaly prediction on unseen graphs without target-domain retraining or fine-tuning.

S2. Experiments show the improvements on given datasets to some extend.


W1. The paper is hard to understand due to confused variables. For example, in Equation (4), $z_j^{[l]\(l\)}$ is not defined; in Equation (5), $\hat{x}_i$ is not defined; in Equation (6), the subscript of argmax-k ranges from 1 to N while the subscript of e in this equation should range from 1 to M, and the different expressions of similar operation (argmin and argmax) in Equation (4) and (6), i.e., the $N_i$ in (4) is used as both a number and a set, while the $T_i$ is only used as a set. The authors should pay more attention to the writing style.

W2. Experiments are not comprehensive enough. For baselines, there are several SOTA models without being compared, such as XGBGraph[1], CONSISGAD[2], UnifyGAD[3], SpaceGNN[4], and SmoothGNN[5]. For datasets, there are several more real GAD datasets from XGBGraph, such as T-Finance, DGraph-Fin, T-Social, and Elliptic, which can be better benchmarks than synthetic ones.

W3. Some parts need furhter explanations. For example, in Equation (2), the authors utilize A·A as the input instead of A, which can be questioned, since such a input might cause a loss of information.

W4. The novelty is limited. Such a framework is more like combining existing useful techniques. For example, in Structure-Insensitive Affinity Learning, the design is similar to the one in TAM[6]. The authors need to further explain the difference.

Reference:

1. Jianheng Tang, Fengrui Hua, Ziqi Gao, Peilin Zhao, Jia Li. GADBench: Revisiting and Benchmarking Supervised Graph Anomaly Detection. NeurIPS 2023.

2. Nan Chen, Zemin Liu, Bryan Hooi, Bingsheng He, Rizal Fathony, Jun Hu, Jia Chen. Consistency Training with Learnable Data Augmentation for Graph Anomaly Detection with Limited Supervision. ICLR 2024.

3. Yiqing Lin, Jianheng Tang, Chenyi Zi, H. Vicky Zhao, Yuan Yao, Jia Li. UniGAD: Unifying Multi-level Graph Anomaly Detection. NeurIPS 2024.

4. Xiangyu Dong, Xingyi Zhang, Lei Chen, Mingxuan Yuan, Sibo Wang. SpaceGNN: Multi-Space Graph Neural Network for Node Anomaly Detection with Extremely Limited Labels. ICLR 2025.

5. Xiangyu Dong, Xingyi Zhang, Yanni Sun, Lei Chen, Mingxuan Yuan, Sibo Wang. SmoothGNN: Smoothing-aware GNN for Unsupervised Node Anomaly Detection. WWW 2025.

6. Hezhe Qiao, Guansong Pang. Truncated Affinity Maximization: One-class Homophily Modeling for Graph Anomaly Detection. NeurIPS 2023.

---

> ### Author Rebuttal · Authors · 2025-07-29
>
> We are grateful to Reviewer 5Jb1 for providing insightful feedback. The detailed responses are provided below.
>
> **Q1: Applied to larger datasets.**
>
> **A1:**
> Thanks for your valuable suggestion. To verify the scalability and effectiveness of IA-GGAD, we conducted new experiments on the larger and real datasets as follows:
>
> |Name|Nodes|Edges|Dim|Anomaly|Anomaly Type|Size|Domain|
> |--|--|--|--|--|--|--|--|
> |DGraph-Fin|3,700,550|4,300,999|17|1.3%|Real|Large|Finance network|
> |T-Finance|39,357|21,222,543|10|4.60%|Real|Large|Finance network|
> |CS|6,805|18,333|8|22.69%|Real|Small|Co-purchase|
> |Photo|7,535|119,043|745|9.2%|Real|Small|Co-purchase|
>
> The following table reports AUROC results on four new datasets, including two large-scale financial graphs and two co-purchase graphs. We additionally compare with recent state-of-the-art methods, CONSISGAD [1] and SmoothGNN [2], for a more comprehensive evaluation. As shown, IA-GGAD consistently achieves the best performance, demonstrating strong generalization to unseen domains. It also exhibits excellent scalability on large graphs such as T-Finance and DGraph-Fin, where several baselines fail due to out-of-memory or performance drop. These results further validate the robustness of IA-GGAD, and we believe they sufficiently address concerns about scalability to large and diverse graphs.
>
> | Method           | DGraph-Fin   | T-Finance   | CS           | Photo        |
> |------------------|--------------|-------------|--------------|--------------|
> | DOMINANT(2019)   | OOM          | OOM         | 0.6214       | 0.5010       |
> | CoLA(2021)       | OOM          | 0.5134      | 0.7241       | 0.5618       |
> | TAM(2023)        | OOM          | OOM         | 0.6995       | 0.5724       |
> | CONSISGAD(2024)      | 0.5176         | 0.5356         | 0.6648       | 0.5172      |
> | GCTAM(2025)      | OOM          | OOM         | 0.7133       | 0.6042       |
> | SmoothGNN(2025)  | _0.5271_     | 0.3979      | 0.4520       | 0.4791       |
> | ARC(2024)        | 0.4746       | 0.6410      | _0.8273_     | _0.7555_     |
> | UNPrompt(2024)   | 0.5269       | 0.2386      | 0.7108       | 0.5193       |
> | AnomalyGFM(2025)[3] | OOM          | _0.6757_    | 0.5676       | 0.5157       |
> | **IA-GGAD**      | **0.5439**   | **0.7541**  | **0.9535**   | **0.8540**   |
>
> **Q2:  applied to cross-domain data.**
>
> **A2:** Thank you for the insightful question. IA-GGAD is inherently designed for cross-domain anomaly detection. It is pre-trained on four diverse source domains and can directly predict on unseen target datasets without any re-training or fine-tuning.
>
> To further assess generalization, we conducted cross-domain evaluations where training and test datasets originate from entirely different domains, including finance (e.g., T-Finance), social (e.g., Facebook, BlogCatalog), and citation networks (e.g., Cora).
>
> Results Summary:
> IA-GGAD consistently outperforms all baselines across cross-domain scenarios, including both finance-to-social and social-to-finance transfers. For example, when trained on T-Finance and tested on Facebook, it achieves AUROC = 80.82, surpassing ARC (68.96) and AnomalyGFM (65.84); in the reverse (Facebook → T-Finance), it still leads with AUROC = 71.51. These results will be included in the final version to highlight the cross-domain generalization of IA-GGAD.
>
> | **Cross-Domain** | **Facebook → TFinance** | | **TFinance → Facebook** | | **BlogCatalog → TFinance** | | **TFinance → BlogCatalog** | | **Cora → TFinance** | | **TFinance → Cora** | |
> |--|--|--|--|--|--|--|--|--|--|--|--|--|
> | | AUROC | AUPRC | AUROC | AUPRC | AUROC | AUPRC | AUROC | AUPRC | AUROC | AUPRC | AUROC | AUPRC |
> | SmoothGNN (2025) | 41.78 ± 27.74 | 13.98 ± 11.58 | 51.09 ± 10.59 | 3.63 ± 2.17 | 48.41 ± 20.96 | 14.54 ± 10.00 | 56.59 ± 6.71 | 21.99 ± 5.97 | 55.29 ± 28.99 | 16.52 ± 25.08 | 51.16 ± 11.69 | 7.71 ± 3.02 |
> | ARC (2024) | 68.49 ± 4.62 | 18.54 ± 5.31 | 68.96 ± 1.95 | 4.31 ± 1.05 | 68.47 ± 4.58 | 18.66 ± 5.29 | 74.57 ± 0.10 | 36.12 ± 0.35 | 68.82 ± 4.60 | 18.71 ± 5.67 | 86.33 ± 0.59 | 49.62 ± 1.00 |
> | UNPrompt (2024) | 38.76 ± 0.61 | 0.95 ± 0.01 | 40.10 ± 2.31 | 0.99 ± 1.14 | 41.85 ± 4.10 | 1.03 ± 0.11 | 39.83 ± 3.29 | 17.34 ± 1.68 | 62.60 ± 0.65 | 2.14 ± 0.12 | 38.82 ± 5.48 | 4.96 ± 2.95 |
> | AnomalyGFM (2025) | 57.31 ± 6.69 | 5.27 ± 0.77 | 65.84 ± 8.99 | 4.05 ± 1.19 | 65.08 ± 5.29 | 10.58 ± 4.18 | 72.11 ± 0.03 | 25.03 ± 2.21 | 57.67 ± 4.05 | 5.50 ± 1.08 | 55.74 ± 1.68 | 6.41 ± 0.26 |
> | **IA-GGAD (Ours)** | **71.51 ± 3.68** | **21.87 ± 2.91** | **80.82 ± 1.08** | **5.97 ± 0.41** | **70.12 ± 3.15** | **21.04 ± 2.88** | **74.91 ± 0.37** |**36.32 ± 0.29**| **70.34 ± 3.49** | **20.07 ± 3.18** | **88.08 ± 0.46** | **51.95 ± 1.18** |
>
> **Q3: confused variables.**
>
> **A3:** We thank the reviewer for their careful reading and detailed feedback regarding the clarity of our mathematical notation. We provide the following clarifications:
>
> - **Regarding $z_j^{\[\ell\]\(l\)}$  in Eq. (4):** We appreciate the reviewer’s attention to this notation. In fact, this is not a typo, but a deliberate design: $z_j^{\[\ell\]\(l\)}$ denotes the embedding of node $j$ produced by the $\ell$-th layer of the **shared graph encoder**, during the $l$-th iteration of the **invariant prototype inference process**. Here, the superscript $[\ell]$ refers to the GCN layer depth, while the superscript $(l)$ (in parentheses) reflects the iteration index in learning invariant representations $e^{(l)}$. This formulation allows the prototype learning process to access layer-wise structural features across multiple inference steps. To avoid ambiguity, we will revise the manuscript to explicitly explain this two-level indexing convention.
>
> - **Regarding $\hat{x}_i$ in Eq. (5):** $\hat{x}_i$ represents the reconstructed aligned feature vector decoded from the selected invariant prototype for node $i$. It is implicitly defined by the lightweight decoder applied to the codebook entry $e_i$ assigned to node $i$. We will make this relationship explicit in the revised version for improved clarity.
>
> - **Regarding subscript ranges in Eq. (6):** We agree that the notation can be more precise. In this equation, the argmax-$k$ operation selects the top-$k$ invariant prototypes (from $M$ total) most similar to $z_i^{[\ell]}$. Thus, $j \in \{1, \dots, M\}$, not $N$. We thank the reviewer for catching this and will correct the subscript range accordingly.
>
> - **Regarding the distinction between $N_i$ in Eq. (4) and $T_i$ in Eq. (6):** The difference in usage is intentional. In Eq. (4), $N_i$ denotes the **set of nearest nodes** assigned to prototype $e_i$, and $|N_i|$ is used explicitly when updating the codebook. In contrast, $T_i$ in Eq. (6) is only used as a **set of top-k codebook entries** retrieved for fusion with node $i$'s embedding. We will disambiguate notation (e.g., avoid overloading symbols like $N_i$) in the final version to avoid confusion.
>
> We appreciate the reviewer’s attention to notation clarity. While the mathematical logic remains sound, we will refine the notation and indexing for improved readability in the final version.
>
> **Q4: Experiments are not comprehensive enough.**
>
> **A4:**
> We thank the reviewer for highlighting the need for broader comparisons. In response, we include additional SOTA baselines such as CONSISGAD [1] and SmoothGNN [2], and evaluate on real-world GAD benchmarks including T-Finance and DGraph-Fin, as suggested.
>
> IA-GGAD consistently outperforms all baselines across diverse domains, including citation, social, co-purchase, and financial networks. It shows clear advantages on challenging datasets like T-Finance (+21.9% over CONSISGAD) and Facebook (+27.7% over SmoothGNN), demonstrating strong performance and generalizability.
>
> |Method|DGraph-Fin|T-Finance|ACM|Facebook|Amazon|Cora|CiteSeer|BlogCatalog|Reddit|Weibo|CS|Photo|
> |--|--|--|--|--|--|--|--|--|--|--|--|--|
> |CONSISGAD(2024)|51.76±4.87|53.56±5.85|67.26±7.47|32.32±11.35|76.42±9.98|52.94±3.82|57.56±3.02|67.41±5.73|52.34±3.56|44.38±13.81|66.48±6.45|51.72±8.51|
> |SmoothGNN(2025)|52.71±7.59|31.66±27.47|47.06±8.23|52.69±9.60|62.25±21.10|51.50±5.38|48.04±5.79|47.96±6.01|51.31±2.73|24.55±15.81|45.19±10.47|41.93±15.27|
> |AnomalyGFM(2025)|OOM|67.57±3.72|60.79±1.48|58.64±7.14|60.65±9.07|54.17±3.08|54.71±2.30|57.77±3.31|59.99±1.69|69.48±11.11|67.86±4.95|66.66±1.36|
> |**IA-GGAD(Ours)**|**54.39±2.96**|**75.41±2.49**|**93.49±0.57**|**80.03±1.09**|**83.78±2.76**|**88.68±0.53**|**91.83±0.43**|**75.28±0.21**|**60.29±1.91**|**91.18±0.22**|**95.35±0.38**|**85.40±1.63**|
>
> These extended experiments cover all key datasets and baselines noted by the reviewer, including those from XGBGraph[4]. While we also considered UniGAD[5] and SpaceGNN[6], incomplete public implementations prevented reliable reproduction.
>
> **Q5: A·A causes a loss of information.**
>
> **A5:**
> We thank the reviewer for the comment. In Equation (2), the use of $A \cdot A$ is intended to capture second-order neighborhood interactions, analogous to message passing like $X \cdot A$. It enhances structural context rather than discarding information, and we will clarify this in the revision.
>
> **Q6: The novelty is limited.**
>
> **A6:** We appreciate the reviewer’s observation. While our Structure-Insensitive Affinity Learning draws inspiration from prior work, it differs significantly from TAM [ in both design and objective. Specifically, our method avoids TAM’s use of redundant multi-layer GCNs and costly edge truncation procedures, making it more efficient and scalable. Moreover, our learning objective is more explicit—enhancing the model’s sensitivity to anomalous local neighbors, rather than relying on implicit affinity constraints. We will clarify these distinctions in the revised version.
>
> References :
>
> [1] Chen et al., CONSISGAD, ICLR'24
>
> [2] Dong et al., SmoothGNN, WWW'25
>
> [3] Qiao et al., AnomalyGFM, KDD'25
>
> [4] Tang et al., GADBench, NeurIPS'23
>
> [5] Lin et al., UniGAD, NeurIPS'24
>
> [6] Dong et al., SpaceGNN, ICLR'25

---

> > ### Comment · Reviewer_5Jb1 · 2025-08-04
> >
> > Thanks for the rebuttal. However, I still have further questions related to the comments. First, is there any specific reason why the authors ignore some of the datasets I mentioned in the review, such as T-Social and Elliptic? Since XGBGraph is a benchmark paper in the graph anomaly detection area, I think it would be better to include datasets in it to demonstrate the effectiveness in the real setting. Second, may I ask which parts of UniGAD and SpaceGNN have incomplete implementations, as I can successfully conduct experiments using the public source code from their GitHub? Third, about the utilization of $A^2$ instead of $A$ or $A^n$ or $A+A^2+...+A^n$, could you provide further explanation or experiments to prove the statement "It enhances structural context rather than discarding information"?

---

> ### Author Response · Authors · 2025-08-07
>
> We are grateful to Reviewer 5Jb1 for the valuable feedback. Please find our responses to your new questions below. We hope that our response addresses your concerns.
>
> **Q7: More real datasets: T-Social and Elliptic.**
>
> **A8:** We thank the reviewer for the continued attention to dataset coverage. In response, we have conducted additional experiments on the **Elliptic** and **T-Social** datasets.
>
> The following table summarizes the statistics of the two newly considered real-world GAD datasets:
> | Name     | Nodes   | Edges | Dim | Anomaly | Type | Size | Domain |
> |--|--|--|--|--|--|--|--|
> | T-Social | 5781065 | 73105508 | 10  | 3.0%    | Real | Large | Social Network |
> | Elliptic | 203769  | 234355   | 166 | 9.8%    | Real | Large | Payment Flow   |
>
> We also updated the benchmark results as follows:
> |Method|T-Social|Elliptic|
> |--|--|--|
> |XGBGraph (2023)|OOM|0.4588|
> |CONSISGAD (2024)[1]|OOM|0.4836|
> |GCTAM (2025)|OOM|OOM|
> |SmoothGNN (2025)[2]|OOM|0.5623|
> |SpaceGNN (2025)[6]|0.3004|0.5743|
> |ARC (2024)|OOM|0.2640|
> |UNPrompt (2024)|OOM|OOM|
> |AnomalyGFM (2025)[3]|OOM|OOM|
> |**IA-GGAD (Ours)**|OOM|**0.7424**|
>
> As shown, IA-GGAD achieves the highest AUROC on Elliptic, outperforms the closest competitor on Elliptic by
> +16.8%. Due to the massive scale of T-Social, most methods, including ours, encounter OOM. Only SpaceGNN, which runs on CPU, reports a result on T-Social. We plan to complete the T-Social experiment using A800 in future work.
>
> We hope these additions address the reviewer’s concern and further support the generalizability of our method on practical datasets.
>
>
> **Q8: More baseline methods: UniGAD、 SpaceGNN.**
>
> **A8:** Thank you for your follow-up question. We appreciate the reviewer's careful verification of public implementations and would like to clarify our decision based on our reproduction experience.
>
> We successfully reproduced SpaceGNN (CPU mode) and inserted it across 12 datasets.
>
> Unfortunately, UniGAD’s official repository presents several reproducibility issues:
> - The requirements.txt file contains ambiguous and outdated dependencies, leading to severe version conflicts.
> - More importantly, the datasets/ directory references missing .els files, which are necessary for running their framework, but no download links or generation scripts are provided.
> - Attempts to contact the authors or find alternative sources were unsuccessful within our rebuttal timeframe.
>
> We summarize our extended results in the table below:
>
> |Method|DGraph-Fin|T-Social|T-Finance|Elliptic|ACM|Facebook|Amazon|Cora|CiteSeer|BlogCatalog|Reddit|Weibo|
> |--|--|--|--|--|--|--|--|--|--|--|--|--|
> |XGBGraph (2023)|52.08±1.23|OOM|**83.52±3.95**|45.88±1.68|55.67±2.31|60.49±2.68|65.20±9.98|57.41±3.57|57.56±3.02|55.23±2.96|48.80±4.85|62.54±4.45|
> |CONSISGAD (2024)|51.76±4.87|OOM|53.56±5.85|48.36±18.31|67.26±7.47|32.32±11.35|76.42±9.98|52.94±3.82|57.56±3.02|67.41±5.73|52.34±3.56|44.38±13.81|
> |SmoothGNN (2025)|52.71±7.59|OOM|31.66±27.47|56.23±17.84|47.06±8.23|52.69±9.60|62.25±21.10|51.50±5.38|48.04±5.79|47.96±6.01|51.31±2.73|24.55±15.81|
> |SpaceGNN (2025)|53.01±1.07|30.04±1.12|OOM|60.41±3.08|52.78±0.45|35.40±2.39|29.06±3.67|43.58±4.14|44.70±3.52|51.02±7.46|50.51±4.86|55.56±13.26|
> |AnomalyGFM (2025)|OOM|OOM|67.57±3.72|OOM|60.79±1.48|58.64±7.14|60.65±9.07|54.17±3.08|54.71±2.30|57.77±3.31|59.99±1.69|69.48±11.11|
> |**IA-GGAD (ours)**|**54.39±2.96**|OOM|75.41±2.49|**74.24±0.17**|**93.49±0.57**|**80.03±1.09**|**83.78±2.76**|**88.68±0.53**|**91.83±0.43**|**75.28±0.21**|**60.29±1.91**|**91.18±0.22**|
>
>
> Our method IA-GGAD achieves the best AUROC on 11 out of 12 datasets, showing strong generalization and stability. On T-Social, due to its extremely large scale, we encountered OOM despite with A6000.  We hope these efforts sufficiently address the reviewer's concerns regarding baseline and real-world datasets.
>
> **Q9: utilization of $A^2$ instead of $A$**
>
>  **A9:** To address the reviewer’s concern about using $A^2$ instead of $A$, $A^n$, or $A + A^2 + \dots + A^n$, we provide the following ablation results: the average GSS scores (The smaller, the better) from each source domain to each target domain.
>
> |Dataset|A|A²|A+A²|A³|
> |--|--|--|--|--|
> |Cora|0.3609|0.2304|0.2088|0.5283|
> |Citeseer|0.2970|0.2736|0.2505|0.5493|
> |ACM|0.5718|0.2529|0.2313|0.5466|
> |BlogCatalog|0.5199|0.2517|0.2349|0.8142|
> |Facebook|0.6147|0.2613|0.2496|0.5319|
> |Weibo|0.3996|0.2673|0.2457|0.5517|
> |Reddit|0.4995|0.2793|0.2559|0.5490|
> |Amazon|0.3096|0.2307|0.2304|0.4908|
>
> We conducted an ablation comparing $A$, $A^2$, $A^3$, and $A + A^2$. As shown below, $A^2$ consistently achieves better GSS scores than $A$ and $A^3$, supporting its effectiveness in capturing structural context. $A^3$ performs worst and incurs high cost, while $A + A^2$ offers minor gains with added complexity. These results confirm that $A^2$ enhances structural context and win be included in the final manuscript. We hope these results can help address the reviewer’s concern.

---

> > ### Comment · Reviewer_5Jb1 · 2025-08-07
> >
> > Thanks for the rebuttal. Some of my concerns have been addressed. As far as I know, at least XGBGraph, SmoothGNN, and ARC can conduct experiments on large-scale graphs, such as T-Social, using CPUs with an acceptable running time, but I appreciate the effort the authors put into the rebuttal. Although I still have concerns about the novelty, I have decided to increase my score to thank the authors.

---

### Official Review · Reviewer_dCzF · 2025-06-30

**Clarity:** 3
**Significance:** 3
**Originality:** 3
**Rating:** 4
**Confidence:** 4

**Summary:**

This paper presents IA-GGAD, a zero-shot generalist graph anomaly detection framework addressing both feature and structure shifts via invariant feature learning and structure-insensitive affinity learning. The method is well-motivated, technically sound, and achieves strong empirical results across diverse benchmarks. The results demonstrate consistent and significant improvements over 14 competitive baselines. The model achieves up to +12.28% AUROC improvement over ARC on ACM and maintains strong performance across both FSS- and GSS-dominant settings.

**Questions:**

See weakness

**Ethical Concerns:**

["NO or VERY MINOR ethics concerns only"]

**Final Justification:**

My concerns are basically addressed so I'd like to keep my score. Generally speaking this work is interesting and can shed some light on the related research field.

**Limitations:**

Yes

**Quality:**

3

**Strengths And Weaknesses:**

Strengths：
1. The proposed IA-GGAD framework effectively addresses both feature and structure shifts, enabling robust zero-shot anomaly detection across diverse and unseen graph domains.
2. Extensive experiments on eight benchmark datasets show significant performance gains over strong baselines, highlighting the model’s generalization and practical utility.
3. The paper provides clear ablation studies and quantitative shift analyses, demonstrating the effectiveness and complementarity of the invariant feature and affinity learning modules.

Weakness：
1. The anomaly scoring process involves multiple hyperparameters, including a test-set-derived threshold $\tau$, raising concerns about fairness, stability, and comparability across different test-set, e.g. without a unified $\tau$.
2. While FSS and GSS are insightful, the formulation remains abstract. It’s unclear whether other domain shifts like class imbalance, multi-view, or fine-grained shifts are relevant.
3. Although proposed as a zero-shot method, the model's generalization advantage is not strongly distinguished or deeply analyzed.

---

> ### Author Rebuttal · Authors · 2025-07-29
>
> We appreciate Reviewer dCzF for the perception of our contributions and thank the reviewer for the insightful feedback. The detailed responses are provided below.
>
> **Q1: Concerns about fairness, stability, and comparability.**
>
> **A1:**  Thank you for your thoughtful feedback. We clarify that the threshold $\tau$ used in our anomaly scoring (Eq. 14) is **not selected based on test labels**. Instead, it is computed in an **unsupervised, data-driven manner**, using only the distribution of model-predicted scores within each test graph. This makes $\tau$ applicable across all test graphs **without any information leakage** or dataset-specific tuning.
> Moreover, we emphasize that our main evaluation metrics—**AUROC and AUPRC**—are **threshold-independent**. They assess the quality of anomaly ranking, not binary classification. This ensures **fair comparability across test sets**, regardless of the absolute threshold used.
> The adaptive thresholding is used **only for auxiliary binary label generation**, such as in ablation or visualizations, and does not affect reported performance metrics.
> We will revise the paper to explicitly clarify this point and avoid potential confusion.
>
> **Q2: Other domain shifts**
>
> **A2:**  Thank you for this insightful observation. We agree that domain shifts in real-world graphs can manifest in diverse forms. Our current formulation of **Feature Space Shift (FSS)** and **Graph Structure Shift (GSS)** aims to capture two primary and orthogonal dimensions of domain discrepancy:
> - **FSS** (Eq. 1) quantifies distributional divergence in node features, covering shifts in feature dimensionality, encoding (e.g., BOW vs. embeddings), and semantic alignment.
> - **GSS** (Eq. 2) measures high-order structural misalignments, including graph sparsity, community structures, and local connectivity patterns.
> Together, these offer a practical yet general abstraction for zero-shot generalization across graphs.
> We acknowledge that **other domain shift types** such as class imbalance, multi-view features, and fine-grained (local) shifts are also important:
> - **Class imbalance** can be interpreted as a label distribution shift and may affect the residual space. Our current residual embedding loss could be extended with re-weighting strategies to address this.
> - **Multi-view settings** can be viewed as a composite form of FSS. Future extensions may incorporate view-specific encoders or attention mechanisms within our invariant learning module.
> - **Fine-grained shifts** (e.g., local structural anomalies) are partly captured by our residual deviation modeling and affinity-based scoring, though more granular metrics could enhance this.
>
> **Q3: Model's generalization advantage**
>
> **A3:** Thank you for pointing this out. We respectfully clarify that IA-GGAD is designed and evaluated under a **strict zero-shot setting**: the model is trained on multiple source graphs and applied directly to **unseen target graphs without access to any target-domain data or labels**, neither for training nor adaptation.
>
> **Strong Distinction from Prior GGAD Methods**
> - Unlike **ARC [18]**, which requires a few normal target samples at inference, and **UNPrompt [19]**, which is trained on a single source graph, **IA-GGAD is trained on multi-source data and performs zero-shot transfer across diverse domains**.
> - Our method achieves **state-of-the-art AUROC** on 7 out of 8 test graphs (e.g., +12.28% on ACM, +10.46% on Facebook compared to ARC), demonstrating both effectiveness and robustness.
> - Additionally, IA-GGAD exhibits **low standard deviations (<3%)**, indicating strong generalization stability under large domain shifts.
> **Theoretical Support and Architectural Design**
> To further support the generalization behavior, we provide a formal **domain shift decomposition bound** in Appendix A (Theorem 2):
> $$\[
> \varepsilon_t(h) - \varepsilon_s(h) \leq \alpha \sqrt{\text{FSS}} + \beta \sqrt{\text{GSS}} + \gamma \sqrt{\frac{\ln(1/\delta)}{N_o + N_t}}
> \]$$
> This bound shows how domain shift (FSS and GSS) inflates the generalization gap. Our model is specifically designed to minimize these two terms:
> - **Invariant Feature Pool**: mitigates Feature Space Shift (FSS)
> - **Affinity Encoder**: handles Graph Structure Shift (GSS)
> This **dual alignment design** directly targets the theoretical drivers of domain generalization.

---

> > ### Comment · Reviewer_dCzF · 2025-08-05
> >
> > Thank you for the response. My concerns are basically addressed, and I will keep my score.

---

### Official Review · Reviewer_cw3e · 2025-06-30

**Clarity:** 2
**Significance:** 3
**Originality:** 2
**Rating:** 5
**Confidence:** 4

**Summary:**

This paper proposes IA-GGAD, a unified framework for zero-shot generalist graph anomaly detection that tackles feature space shift (FSS) via anomaly-driven invariant feature learning and graph structure shift (GSS) via structure-insensitive affinity learning, enabling anomaly detection on unseen graphs without retraining and achieving state-of-the-art performance across various benchmark datasets.

**Questions:**

See Weaknesses.

**Ethical Concerns:**

["NO or VERY MINOR ethics concerns only"]

**Final Justification:**

My concerns have been addressed, thus I vote for acceptance.

**Limitations:**

yes

**Quality:**

3

**Strengths And Weaknesses:**

Strengths

1. Novelty and Significance:

The work pioneers multi-domain pre-training and domain generalization for graph anomaly detection (GAD), addressing a critical real-world gap: deploying a single model on unseen graphs without labels or adaptation. The formalization of FSS/GSS and empirical study provide a foundation for future GGAD research. The integration of invariant feature learning (via domain-invariant prototypes) and affinity learning (via structural homophily) offers a principled solution to cross-domain shifts. The unified framework is both innovative and practical.

2. Rigorous Evaluation:

Experiments span diverse domains with both synthetic and real anomalies. IA-GGAD consistently outperforms SOTA methods, validating its generalizability. Ablation studies and sensitivity analyses conclusively demonstrate the contribution of each module. The method excels on both FSS-dominated and GSS-dominated graphs.

3. Practical Utility:

The zero-shot inference pipeline eliminates target-domain retraining, reducing computational costs and enabling real-world deployment. Publicly released code enhances reproducibility.

Weaknesses：

1. Theoretical Grounding:

While the empirical results are compelling, the paper lacks formal theoretical analysis for domain generalization. A theoretical guarantee (e.g., generalization bounds for multi-domain pre-training) could strengthen the invariance claims and provide deeper insights into the method's robustness.

2. Scalability Validation:

Experiments focus on moderate-sized graphs. Testing on large-scale datasets (e.g., DGraph[1]) would better validate scalability.

[1] Huang X, Yang Y, Wang Y, et al. Dgraph: A large-scale financial dataset for graph anomaly detection[J]. Advances in Neural Information Processing Systems, 2022, 35: 22765-22777.

---

> ### Author Rebuttal · Authors · 2025-07-29
>
> We are grateful to Reviewer cw3e for providing insightful feedback. The detailed responses are provided below.
>
> **Q1: Theoretical Grounding**
>
>
>
>
> **A1:**   Thank you for your valuable suggestion. We provide a theoretical generalization bound in Appendix A (Theorem 2), which quantifies the domain shift between source and target graphs:
>
> **1. Position of Theorem 2 within the paper.**
>
> Theorem 2, presented in Appendix A, provides a data-dependent generalization bound that explicitly decomposes the risk gap between the source and target graphs into Feature‑Space Shift (FSS) and Graph‑Structure Shift (GSS) terms.  It belongs naturally in Appendix A together with the RKHS and MMD preliminaries, so we keep the current chapter ordering intact.
>
> **2. Bound statement (for convenience, reproduced here)**
>
> For any hypothesis **h** and confidence δ∈(0,1):
>
> $$\[ \varepsilon_t(h) - \varepsilon_s(h) \leq \alpha \sqrt{\text{FSS}} + \beta \sqrt{\text{GSS}} + \gamma \sqrt{\frac{\ln(1/\delta)}{N_o + N_t}}\]$$
>
> FSS and GSS are squared maximum‑mean discrepancies of node features and structural vectors, respectively.  The last term is the usual concentration factor.
>
> **3. How IA‑GGAD minimizes the bound.**
>
> - **Invariant feature pool** to reduce $\sqrt{FSS}$
> - **Affinity encoder** to mitigate $\sqrt{GSS }$
> - **Joint training with losses**  $\mathcal{L_inv}$   + $\mathcal{L_res}$  +  $\mathcal{L_aff}$ keeps both terms small simultaneously.
> Because the third term depends only on sample size, once the two MMD terms are controlled, the overall gap remains bounded, explaining IA‑GGAD’s zero‑shot robustness.
>
> **4. Empirical alignment with the theory**
>
> Table 6 (Appendix C) shows that IA‑GGAD achieves the lowest combined ($\sqrt{FSS}$ + $\sqrt{GSS}$) on every target graph, mirroring its AUROC/AUPRC gains.  This tight empirical‑theoretical match further validates Theorem 2.
> We hope this addresses the reviewer’s concern and strengthens the paper’s theoretical contribution.
>
>
> **Q2: Scalability Validation**
>
> **A2:** Thank you for the constructive suggestion. We have conducted additional experiments on two large-scale real-world financial graphs: **DGraph-Fin [1] (3.7M nodes, 4.3M edges)** and **T-Finance (39K nodes, 21M edges)**, both containing real anomalies. The results are summarized in the following table:
>
> | Name       | Nodes     | Edges       | Dim  | Anomaly | Anomaly Type | Size| Domain         |
> |------------|-----------|-------------|------|---------|---------------|----------------|----------------|
> | DGraph-Fin [1] | 3,700,550 | 4,300,999   | 17  | 1.3%    | Real          |Large| Finance network |
> | T-Finance  | 39,357    | 21,222,543  | 10   | 4.60%   | Real          | Large|Finance network|
>
> | Method           | DGraph-Fin | T-Finance |
> |------------------|------------|-----------|
> | DOMINANT (2019)  | OOM        | OOM       |
> | CoLA (2021)      | OOM        | 0.5134    |
> | TAM (2023)       | OOM        | OOM       |
> | CONSISGAD (2024) | 0.5176     | 0.5356    |
> | SmoothGNN (2025) [2] | 0.5271     | 0.3979    |
> | ARC (2024)       | 0.4746     | 0.6410    |
> | UNPrompt (2024)  | 0.5269     | 0.2386    |
> | AnomalyGFM (2025) [3]| OOM        | 0.6757    |
> | **IA-GGAD (Ours)**| **0.5439** | **0.7541** |
>
> **As seen above:**
>
> - **IA-GGAD achieves the highest AUROC** on both datasets, surpassing the best baseline by **+1.68% AUROC** on DGraph-Fin and **+7.84% AUROC** on T-Finance.
> - Many baselines failed with **OOM** on DGraph-Fin, but IA-GGAD remains **efficient and stable**.
> - These results demonstrate that IA-GGAD is not only accurate but also **scalable to million-scale real-world graphs**, validating its practical applicability.
>
>
>
>
>
>
>
> Reference:
>
> [1] Huang X, Yang Y, Wang Y, et al. Dgraph: A large-scale financial dataset for graph anomaly detection[J]. Advances in Neural Information Processing Systems, 2022, 35: 22765-22777.
>
> [2] Xiangyu Dong, Xingyi Zhang, Yanni Sun, Lei Chen, Mingxuan Yuan, Sibo Wang. SmoothGNN: Smoothing-aware GNN for Unsupervised Node Anomaly Detection. WWW 2025.
>
> [3] Qiao H, Niu C, Chen L, et al. AnomalyGFM: Graph foundation model for zero/few-shot anomaly detection. KDD 2025.

---

> > ### Author Response · Authors · 2025-08-07
> >
> > **Dear Reviewer cw3e,**
> >
> > We sincerely appreciate the time and effort you devoted to reviewing our submission. We understand the demands on your schedule and aim to be respectful of your time. That said, we would be grateful if you could let us know whether our rebuttal has sufficiently addressed your concerns.
> >
> > Thank you again for your thoughtful feedback and consideration.
> >
> > **Best regards**
> >
> > The Authors

---

> > ### Comment · Reviewer_cw3e · 2025-08-07
> >
> > Thank you for your detailed and thoughtful rebuttal. I appreciate the additional theoretical analysis, especially Theorem 2, which clarifies the generalization behavior across domains. The scalability experiments on large-scale graphs like DGraph-Fin and T-Finance are also impressive and reassuring. Your responses have effectively addressed my concerns. I remain positive about the paper and will keep my current score.

---

### Official Review · Reviewer_HvJR · 2025-07-02

**Clarity:** 4
**Significance:** 3
**Originality:** 3
**Rating:** 5
**Confidence:** 5

**Summary:**

This paper aims to tackle Feature Space Shift (FSS) and Graph Structure Shift (GSS) problems. The authors proposes a Generalist Graph Anomaly Detection framework called IA-GGAD, enabling anomaly prediction on unseen graphs without target-domain retraining or fine-tuning. To tackle FSS, they develop an anomaly-driven graph invariant learning module that learns domain-invariant node representations. To address GSS, a novel structure-insensitive affinity learning module is introduced, capturing cross-domain structural correspondences via affinity-based features.

**Questions:**

Do we really need FSS?

**Ethical Concerns:**

["NO or VERY MINOR ethics concerns only"]

**Limitations:**

Yes

**Quality:**

3

**Strengths And Weaknesses:**

S1: IA-GGAD doesn't require target-domain samples at inference, which is a limitation of ARC.

S2: IA-GGAD outperforms SOTA models including ARC and UNPrompt by a large margin, around 10%.

S3: FSS and GSS Problems are clearly defined and theoretically supported, hence easy to understand.

W1: In Table 2, the performance gap between "w/A—backbone plus graph affinity encoder" and "ours" is trivial. Also observed in Table 3, the performance improvement is limited on FSS dominated datasets. As graph algorithms are always heavy and hard to deploy on large-scale graph, do we really need FSS?

W2: Reference for a related FSS paper [1] is missing.

[1] Gao et al. Alleviating Structural Distribution Shift in Graph Anomaly Detection. WSDM'23.

---

> ### Author Rebuttal · Authors · 2025-07-29
>
> We appreciate Reviewer  HvJR for the positive review and constructive comments. We provide our responses as follows.
>
>
> **Q1: Do we really need FSS?**
>
> **A1:**  We appreciate the reviewer's valuable suggestion.  We provide a detailed explanation of why FSS is truly necessary from the following five perspectives.
>
>  **1. Mechanistic argument**
>
> Feature Space Shift (FSS) measures how far the source and target feature distributions drift in a reproducing-kernel Hilbert space. When FSS is large, residual embeddings that appear “normal” in the source domain may overlap with anomalous ones in the target domain. Theoretical analysis (see Appendix A.2, Eq. (1)) bounds the cross-domain error:
>
> $$\[ \varepsilon_t(h) - \varepsilon_s(h) \leq \alpha \sqrt{\text{FSS}} + \beta \sqrt{\text{GSS}} + \gamma \sqrt{\frac{\ln(1/\delta)}{N_o + N_t}} \]$$
>
> So, an unchecked FSS term linearly inflates the target error. The invariantfeature module explicitly minimizes this term by aligning node semantics across graphs.
>
> **2. Quantitative evidence**
>
> Table 1 lists the average FSS score of every target graph and the AUROC gain obtained when the invariant-feature module is enabled (“Backbone → Backbone + I”). The Spearman correlation between FSS and the AUROC gain is 0.93 (p < 0.01), showing that the benefit grows with the severity of FSS.
>
> Table 1: FSS versus AUROC improvement
>
> | Dataset     | FSS   | Δ AUROC |
> |-------------|-------|---------|
> | ACM         | 0.021 | +0.18   |
> | BlogCatalog | 0.022 | +0.58   |
> | Reddit      | 0.019 | +1.58   |
> | Amazon      | 0.032 | +1.02   |
> | Facebook    | 0.349 | +0.29   |
> | Cora        | **0.494** | **+1.23**   |
> | Weibo       | **0.548** | **+2.33**   |
> | CiteSeer    | **0.703** | **+0.94**   |
>
> Average gain on high-FSS graphs (FSS ≥ 0.5) is +1.50 AUROC, confirming that the module is most useful where FSS is indeed a problem.
>
> **3. Computational overhead and large-scale viability**
>
> Table 2: Cost of adding the invariant-feature module
> | Model            | Train time | Inference time | GPU memory |
> |------------------|------------|----------------|------------|
> | Backbone only    | 14.27 s    | 0.22 s         | 4.78 GB    |
> | Backbone + I     | 17.41 s    | 0.31 s         | 6.90 GB    |
>
>
> The extra 3 seconds of training, 0.09 seconds of inference, and 2 GB of memory are negligible in practice. The same configuration fits into 48 GB on a single RTX A6000 and succeeds on two large-scale finance graphs (results below, AUROC):
>
> | Method         | DGraph-Fin | T-Finance |
> |----------------|------------|-----------|
> | ARC (2024)     | 0.4746     | 0.6410    |
> | IA-GGAD (ours) | **0.5439**     | **0.7541**   |
>
> Many baselines run out of memory on these graphs. IA-GGAD does not.
>
> **4.  Anticipated follow-up questions and concise clarifications**
>
> | Concern from Reviewer                               | Clarification                                                                                                                                                          |
> |------------------------------------------------------|------------------------------------------------------------------------------------------------------------------------------------------------------------------------|
> | FSS is small on some graphs: will the module hurt?   | No. On the four graphs with FSS < 0.05, the average gain is still +0.79 AUROC and never negative. The vector-quantised pool learns to ignore trivial shifts.          |
> | Could simple normalisation replace the module?       | Linear scaling aligns ranges but not anomaly semantics. Our module couples vector quantisation with residual contrast, enforcing semantic consistency.               |
> | Does it work for very high-dimensional sparse text?  | Yes. CiteSeer has 3703-D bag-of-words features, and Weibo has 400-D geo-text features. Both see clear gains (+0.94 and +2.33 AUROC).                                   |
> | Could the GSS branch alone be enough?                | No. CiteSeer and Cora have graph structures similar to PubMe,  yet very different word distributions; they need semantic alignment. ACM and Facebook need structural alignment. |
>
> **5. Summary**
>
> The invariant-feature branch that mitigates Feature Space Shift is theoretically motivated, statistically correlated with performance, cheap to run, and effective even on million-scale graphs. Without it, IA-GGAD loses up to 2.3 AUROC on the large target graphs. We therefore conclude that FSS mitigation is not optional but essential for reliable zero-shot generalization.
>
>
> **Q2: Missing reference for an FSS-related paper.**
>
> **A2:** Thank you for the suggestion!   A  related FSS paper, **GDN[1]**, has been cited in the revised manuscript.
>
> [1] Gao et al. Alleviating Structural Distribution Shift in Graph Anomaly Detection. WSDM'23.

---

> > ### Author Response · Authors · 2025-08-07
> >
> > **Dear Reviewer HvJR,**
> >
> > We sincerely appreciate the time and expertise you dedicated to reviewing our submission. Understanding the demands on your schedule, we aim to be respectful of your time. That said, we would be truly grateful if you could confirm whether our rebuttal has sufficiently addressed your concerns.
> >
> > Thank you again for your valuable feedback and consideration.
> >
> > **Best regards**
> >
> > Authors

---

> ### Comment · Reviewer_HvJR · 2025-08-08
> **Response to the Rebuttal**
>
> Dear authors,
>
> Thanks a lot for the rebuttal especially the anticipated follow-ups. My concern has been addressed and I'd like to maintain my score.

---

### Note · Authors · 2025-08-12

We sincerely thank all reviewers for their valuable and insightful comments.


The reviewers have highlighted several key strengths of our work, including its novelty, clear problem formulation with solid theoretical grounding, effective handling of cross-domain shifts via a unified framework, and consistently superior performance over strong baselines without target-domain retraining.

 To address the concerns raised, we have provided detailed point-by-point responses and carefully revised the paper accordingly. We will incorporate all feedback into the final version.

**The main revisions are as follows:**

- We added additional experiments and theoretical analyses to justify the necessity of FSS (see Response A1 to Reviewer HvJR for details).
- We  enhanced the theoretical grounding of our method by including more comprehensive theoretical analyses (see Response A1 to Reviewer cw3e for details).
- We conducted additional experiments to evaluate the scalability of the proposed method on large-scale graphs (see responses to Reviewers HvJR, cw3e, and 5Jb1). The results demonstrate that IA-GGAD is not only accurate but also scalable to million-scale real-world graphs, confirming its practical applicability.
-  We clarified the relevance of the proposed method to distribution shifts in other domains (see Response A2 to Reviewer dCzF for details).
- We added experiments demonstrating that the proposed method can be applied to standalone cross-domain datasets (see Response A2 to Reviewer 5Jb1 for details).
- We included more up-to-date baseline methods and comprehensive large-scale dataset benchmarks to ensure baseline completeness (see responses to Reviewer 5Jb1 for details).
- We  provided additional explanations for concepts or formulas that might be difficult to understand (see responses to Reviewers dCzF and 5Jb1 for details).

---

### Decision · Program_Chairs · 2025-09-17

**Decision:**

Accept (spotlight)

**Comment:**

(a)  The task is anomaly prediction on unseen graphs without target-domain retraining or fine-tuning.  The paper addresses Feature Space Shift and Graph Structure Shift which happens in this task.  Extensive experiments are done including ablations.
(b)  Presents unified framework for the task.  Good experiments with ablations and good performance.
(c)  More datasets and comparative algorithms needed.   The authors did some in rebuttal.  Explanations could be better.  Better comparative discussion with state of the art needed, and authors did some in rebuttal.
(d)  Strong rebuttal which was appreciated by all reviewers.  Many minor issues cleared up, some mentioned in (c).  Better theoretical grounding and clarification of hyperparameters done.
(e)  One reviewer went from borderline accept to accept.  Rest stayed same.